# Causal manipulation of functional connectivity in a specific neural pathway during behaviour and at rest

Vanessa M Johnen[1][*][†], Franz-Xaver Neubert[1][†], Ethan R Buch[2,3], Lennart Verhagen[1,4], Jill X O'Reilly[5], Rogier B Mars[1,4,5], Matthew F S Rushworth[1,5]

[1]Department of Experimental Psychology, Oxford University, Oxford, United Kingdom; [2]Human Cortical Physiology and Neurorehabilitation Section, National Institute of Neurological Disorders and Stroke, National Institutes of Health, Bethesda, United States; [3]Center for Neuroscience and Regenerative Medicine, Uniformed Services University of Health Sciences, Bethesda, United States; [4]Donders Institute for Brain, Cognition and Behaviour, Radboud University Nijmegen, Nijmegen, Netherlands; [5]Centre for Functional Magnetic Resonance Imaging of the Brain, John Radcliffe Hospital, Oxford University, Oxford, United Kingdom

**\*For correspondence:** vanessa.
johnen@psy.ox.ac.uk

[†]These authors contributed
equally to this work

**Competing interests:** The
authors declare that no
competing interests exist.

**Reviewing editor**: Jody C
Culham, University of Western
Ontario, Canada

**Abstract** Correlations in brain activity between two areas (functional connectivity) have been shown to relate to their underlying structural connections. We examine the possibility that functional connectivity also reflects short-term changes in synaptic efficacy. We demonstrate that paired transcranial magnetic stimulation (TMS) near ventral premotor cortex (PMv) and primary motor cortex (M1) with a short 8-ms inter-pulse interval evoking synchronous pre- and post-synaptic activity and which strengthens interregional connectivity between the two areas in a pattern consistent with Hebbian plasticity, leads to increased functional connectivity between PMv and M1 as measured with functional magnetic resonance imaging (fMRI). Moreover, we show that strengthening connectivity between these nodes has effects on a wider network of areas, such as decreasing coupling in a parallel motor programming stream. A control experiment revealed that identical TMS pulses at identical frequencies caused no change in fMRI-measured functional connectivity when the inter-pulse-interval was too long for Hebbian-like plasticity.

## Introduction

Temporal correlations in activity between brain areas can be measured with functional magnetic resonance imaging (fMRI) and are often referred to as indices of 'functional connectivity' (*Friston, 1994*). Statistical dependencies between remote cortical regions exist both in the absence of external stimuli or task demands (i.e., during the resting state) and during execution of a task (*Hampson et al., 2002*). In the following, the term functional connectivity therefore simply describes a statistical relationship of neural elements with each other. In addition to providing insights into the basic anatomical and physiological organization of healthy neural networks (*Fox and Raichle, 2007*; *Bullmore and Sporns, 2009*; *Smith et al., 2009*; *O'Reilly et al., 2013*), functional connectivity has been used to identify pathological changes occurring in neural circuits in conditions such as stroke (*Wang et al., 2010*), traumatic brain injury (*Bonnelle et al., 2011*; *Ham and Sharp, 2012*), neurodegeneration (*Seeley et al., 2009*), or psychiatric disorders (*Bassett et al., 2008*).

**eLife digest** When a person has their brain scanned, the resulting images show that regions with similar roles tend to be active at the same time. These coordinated patterns of activity are often altered in the brains of patients with neurological or psychiatric disorders. However, relatively little is known about how the patterns are generated.

The degree to which brain regions are active at the same time is thought to depend partly on how well they are connected by brain cells. However, it is also possible that the coordinated activity reflects the extent to which one brain region is able to influence the activity of another. More than 50 years ago, it was demonstrated that this is the case between individual brain cells. If one brain cell repeatedly helps to activate another, the connection between the two cells will be strengthened. This process—known as synaptic plasticity—is thought to support learning and memory.

Now, Johnen, Neubert et al. have shown that the same process can also act between different brain regions. A technique called transcranial magnetic stimulation—in which magnetic fields are applied to specific areas of the scalp to excite brain tissue—was used on human volunteers to activate two regions involved in producing grasping movements with their hands.

If the first region of the brain was repeatedly activated a few milliseconds before the second region as the volunteers reached towards objects, the ability of the first region to activate the second increased. Notably, the effect was not seen when the interval between the activation of the regions was increased to 500 milliseconds: a delay long enough to ensure that brain cells in the first region were no longer active when the second region was stimulated.

This suggests that coordinated changes in the activity of brain regions might reflect the same plasticity processes as changes in activity seen between individual brain cells. This finding raises the possibility that, by deliberately altering the degree of coordinated activity between specific brain regions, it might be possible to recover abilities that have been lost as a result of disorders such as stroke.

Although changes in fMRI-based functional connectivity can be highly specific, their underlying biological mechanisms are less clear. It is thought that functional connectivity patterns are shaped largely by the relatively stable underlying skeleton of structural connections (*O'Reilly et al., 2013*). However, modifications in functional connectivity might also be influenced by changes in synaptic efficacy, for example through changes in the quantity of neurotransmitter release, changes in astrocytes or dendritic spine stabilization.

Here, we aimed to elucidate the contribution of changes in short-term synaptic efficacy to fMRI-based functional connectivity. To this aim, we modulated synaptic efficacy in a specific corticocortical pathway using repetitive paired pulses of transcranial magnetic stimulation (TMS) with a brief inter-pulse interval (IPI; 8 ms) that evoked synchronous pre- and post-synaptic activity and monitored whether those changes were reflected in altered functional connectivity (Experiment 1) (*Figure 1A*). Several TMS protocols have been shown to induce changes in excitability in primary motor cortex (M1) using repetitive stimulation of M1 itself (*Chen et al., 1997*) or stimulation of premotor regions projecting to M1 (*Munchau et al., 2002*). These changes are often thought to reflect frequency-dependent potentiation of synaptic transmission. Furthermore, it has been shown that repetitive paired stimulation of an input into M1—such as the median nerve—and then of M1 itself can change M1 cortico-spinal excitability (*Stefan et al., 2000*; *Wolters et al., 2003*). These paired associative stimulation (PAS) protocols are based upon Hebbian principles of synaptic plasticity and appear to modify connectivity in a controlled manner. Investigations that applied paired-pulse TMS over interconnected sites—for example, homotopical M1 sites (*Rizzo et al., 2009*), M1 and the supplementary motor area (SMA) (*Arai et al., 2011*), and M1 and posterior parietal cortex (*Koch et al., 2013*)—demonstrated altered motor cortical excitability. Notably, the current protocol of repetitive paired-pulse TMS has been shown to induce a causal and directional change of influence of the first brain region (ventral premotor cortex: PMv) over the anatomically connected second region (M1) (*Buch et al., 2011*). This is important since it is such pathway-specific changes that occur in animal models of synaptic plasticity (*Markram et al., 1997*; *Jackson et al., 2006*) and these are

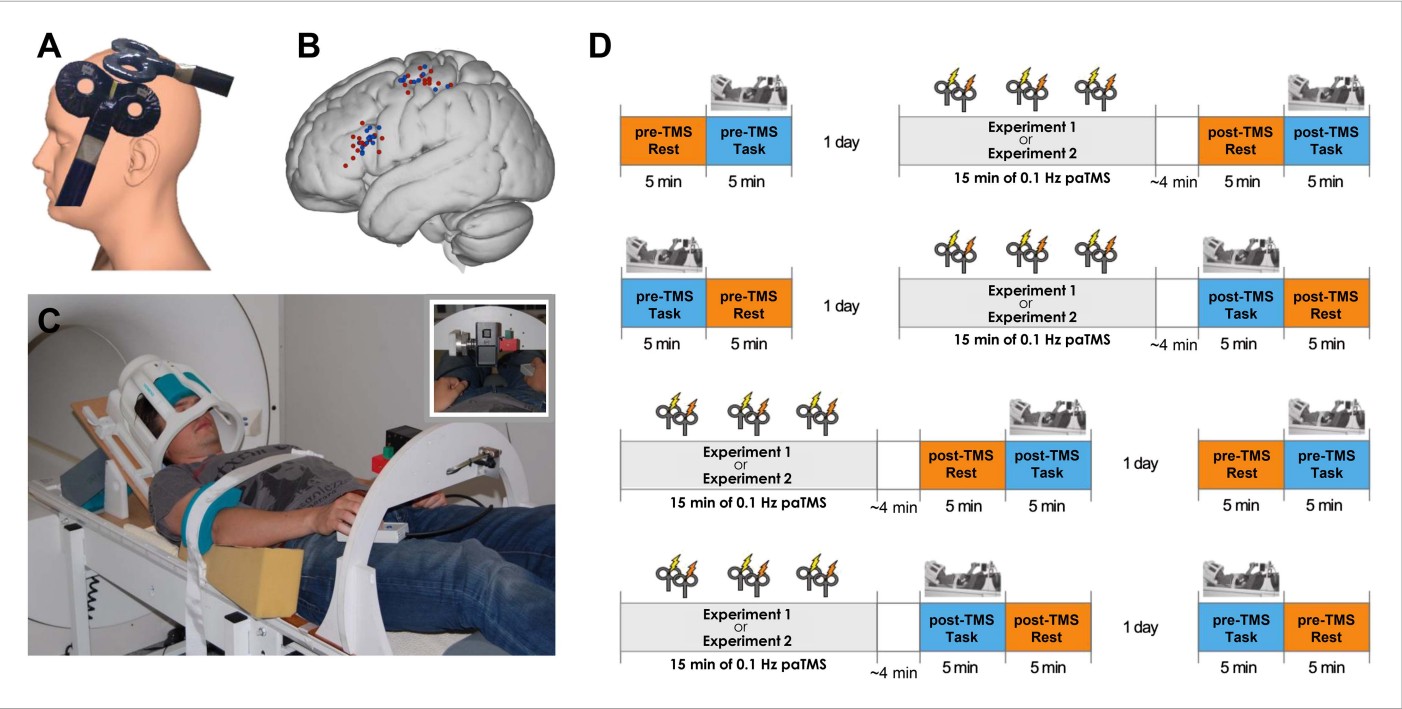

**Figure 1**. Experimental procedure. (**A**) 90 paired pulses were applied over ventral frontal cortex near PMv and M1 (mean MNI coordinates [−56 19 19] and [−40 −18 59] respectively) at 0.1 Hz for 15 min. (**B**) Individual stimulation locations for 8 ms IPI (red) and 500 ms IPI (blue). (**C**) Participants performed visually guided grasping movements towards one of two objects (small or large; see inset) while lying supine in the MR scanner. The head coil was tilted forward by 30° to allow for direct line of sight of the objects to be grasped. A response button box was positioned on the upper leg. (**D**) Experimental design and setup for all experiments (for both 8 ms IPI and 500 ms IPI experiment). The order of resting-state and grasping task fMRI as well as of pre-TMS (baseline) and post-TMS sessions was counterbalanced.

argued to underlie the self-organization proposed to occur in mono-synaptically connected networks in response to regularly occurring input (*Sussillo and Abbott, 2009*).

PMv and M1 are a part of the so-called 'dorsolateral circuit' of areas composed of the anterior intraparietal (AIP) area, areas PF and PFG in the inferior parietal lobule, and PMv and M1 in the frontal lobes. During complex motor behaviour such as reaching and grasping this dorsolateral sensorimotor stream is complemented by a 'dorsomedial circuit' composed of dorsal premotor (PMd), medial intraparietal area (MIP), and posterior superior parietal cortex (pSPL) (*Jeannerod et al., 1995*; *Wise et al., 1997*; *Tanne-Gariepy et al., 2002*; *Galletti et al., 2003*; *Brochier and Umiltà, 2007*; *Grafton, 2010*; *Turella and Lingnau, 2014*).

As is the case for other inter-regional connections, the connections between premotor cortex and M1 are glutamatergic, excitatory ones, but there are synapses on both pyramidal neurons and inhibitory interneurons within M1 (*Tokuno and Nambu, 2000*). This means that although paired stimulation of PMv and M1 leads to strengthening of the excitatory connections between PMv and M1, such strengthening can lead to both enhanced facilitatory and enhanced inhibitory influences of PMv on M1. Enhanced facilitatory influences are more apparent when subjects are subsequently tested while performing a simple reaching and grasping task, and enhanced inhibitory influences are more apparent when subjects are subsequently tested while at rest (*Buch et al., 2011*). These different effects appear as a function of the subject's behavioural and cognitive state at the time of testing (*Bäumer et al., 2009*; *Buch et al., 2010*), but they have not been shown to depend on the subject's cognitive state at the time of plasticity induction (*Buch et al., 2011*). PMv microwire stimulation in macaques has also been shown to exert both facilitatory and inhibitory effects on corticospinal outputs as a function of the animal's state (*Prabhu et al., 2009*).

In the following, this causal and directional influence as quantified by motor-evoked potentials (MEPs) is referred to as 'effective connectivity' (*Friston, 1994*). Classic Hebbian synaptic learning rules such as pathway specificity, spike timing dependency, rapid evolution, persistence for several hours,

and reversibility have been demonstrated for this directed pathway manipulation. Using TMS in this way entails a direct and specific inter-areal manipulation distinct from the compensatory plasticity that occurs following single-site manipulations of neural activity by means of TMS (*Lee et al., 2003*; *O'Shea et al., 2007*; *Grefkes et al., 2010*; *Hartwigsen et al., 2012*).

Recent studies have shown that lesions and disruption of brain areas as well as lesions to connections between brain areas can affect distant areas and connections (*O'Shea et al., 2007*; *Hartwigsen et al., 2012*; *O'Reilly et al., 2013*). These changes are thought to be partly compensatory. For example, in the study by *O'Shea et al. (2007)*, it is suggested that 'activity' in contralateral 'non-dominant' PMd is increased after interruption of ipsilateral PMd. This enhancement of contralateral PMd is accompanied by preserved performance in a stimulus-response matching task. Similarly, *Hartwigsen et al. (2012)* show that action reprogramming can be preserved after PMd interference if the supramarginal gyrus is uncompromised. This study suggests a rapid redistribution of functional weights in order to compensate for interference. Moreover, it has been shown that the interruption of specific pathways has effects far beyond the regions that are directly connected by the pathway (*O'Reilly et al., 2013*). Here by contrast, we study the functional enhancement of a pathway, rather than the disruption of a region or pathway, and its effect on coupling within and outside the targeted network.

To ensure that the changes in functional connectivity we observed could be attributed to plasticity induction, we performed a control experiment of paired-pulse TMS over the same cortical regions (Experiment 2) stimulating with the same number of pulses at the same frequency but with an IPI which precluded spike timing-dependent plasticity (STDP) (IPI: 500 ms). We decided on a 500 ms IPI for the control condition following the exclusion of several other alternative IPIs; we decided against reversing the order of conditioning and test stimulus because we have demonstrated in a previous study that this stimulation order leads to long-term depression-like effects (as assessed by examining the impact of further PMv TMS pulses on M1 (*Buch et al., 2011*); against stimulating both areas at the same time because I-wave interactions may occur at such IPIs (*Prabhu et al., 2009*); against any time interval below 50 ms because there is evidence of plasticity induction at such intervals within the motor system of freely behaving monkeys (*Jackson et al., 2006*). Moreover, we noted that long-interval intracortical inhibition (LICI) within M1 has been demonstrated with TMS using IPIs of up to 200 ms (*Valls-Solé et al., 1992*). Admittedly, other intervals in the hundred milliseconds range might equally have been chosen. Targeting the same cortical areas controlled for the impact of stimulation on brain activity in each component node that the pathway interconnects; if changes in connectivity are simply attributable to stimulation of each area, rather than increased pathway efficacy, then changes in functional connectivity ought to be comparable in Experiment 1 and 2. Here, we show that pathway functional connectivity was not modulated in Experiment 2.

By increasing synaptic efficacy in a corticocortical connection—PMv-M1—involved in complex motor behaviour, we were able to study the relationship of induced plasticity and functional connectivity during the performance of a motor task as well as during the resting state. Further, investigation of a wider motor network provided information about functional reorganisation in response to pathway-specific plasticity induction.

In Experiment 3, we directly tested whether estimates of pathway connectivity based on fMRI data share construct validity with measures of effective connectivity indexed by MEP amplitude ratios across subjects. We demonstrated a direct correlation between the strengths of the two measures for both cognitive states (i.e., resting and grasping), however, the sign of net effective influence of one neural node over another was only determined by TMS-evoked measures and not by fMRI functional connectivity.

## Results

In Experiment 1, each participant (N = 15) underwent two sets of two 5-min fMRI scans for the purpose of assessing functional connectivity in both a baseline state and then again immediately after application of repeated paired-pulse TMS to both PMv and M1 (90 pulse pairs; repeated at 0.1 Hz for 15 min; *Figure 1A,B*). The connections from premotor cortex to M1 are excitatory glutamatergic ones (*Tokuno and Nambu, 2000*), but within M1 they synapse on both excitatory pyramidal neurons and inhibitory interneurons. Such PMv TMS pulses therefore induce a combination of facilitatory and inhibitory influences on M1 activity; which influence becomes most visible depends on the TMS pulse intensity and subjects' cognitive state at the time of testing (i.e., following plasticity induction)

(*Bäumer et al., 2009*; *Buch et al., 2010*, *2011*). We counterbalanced the order of baseline and post-TMS scans across subjects (half of the subjects had the post-TMS scan before the baseline scan on two different days; *Figure 1D*).

Placement of the frontal TMS coil, determined on the basis of sulcal anatomy, was just ventral to the convergence of the inferior precentral sulcus and inferior frontal sulcus and therefore at the border of the dysgranular premotor–prefrontal transition area, the inferior frontal junction (IFJ) and ventral premotor area 6v (*Neubert et al., 2014*). For brevity, we refer to the area as PMv.

We chose to examine this pathway because it is better understood than many and can be investigated in a number of ways even in humans; in monkeys PMv provides one of the principal inputs into M1, and it exerts a powerful influence over M1 output (*Shimazu et al., 2004*; *Dum and Strick, 2005*), and in humans it has been established that paired-pulse TMS of PMv-M1 at an 8 ms IPI, or at similar IPIs, modulates the behavioural and electromyographic (EMG) consequences that are normally observed when M1 is stimulated alone (*Davare et al., 2008*, *2009*; *Buch et al., 2010*; *Neubert et al., 2010*; *Buch et al., 2011*). Moreover, the broader functional circuits with which the pathway is related have been characterized both at rest (*Neubert et al., 2014*) and during simple motor tasks (*Grol et al., 2007*).

The modulating influence of PMv over M1 changes depending on whether subjects are at rest or engaged in different types of motor tasks during application of the TMS pulses (*Davare et al., 2008*, *2009*; *Buch et al., 2010*; *Neubert et al., 2010*; *Buch et al., 2011*). This suggests that the characteristic connectivity modes within the stimulated connection are related to different cognitive states. We therefore probed whether repeated paired-pulse TMS affected functional connectivity in this pathway differently during resting state or prehension performance.

During the scans, participants were instructed either to be at rest or repeatedly to make prehension movements towards one of two objects of different sizes (task; *Figure 1C*). Each rest and task period lasted 5 min and during each period one fMRI scan was acquired. This reaching-and-grasping task is known to produce activity in a network of motor and motor association areas including M1 and PMv and also in AIP (*Grol et al., 2007*). Together these areas, AIP-PMv-M1, constitute a dorsolateral sensorimotor circuit. Another parallel parieto-frontal circuit—the dorsome-dial sensorimotor circuit—has also been linked to the control of reaching and grasping and consists of more dorsal areas, including PMd and parietal area pSPL. Visual input into the two parallel visuomotor circuits during the task is provided via area V3A. *Grol et al. (2007)* assessed connectivity between these same parietal and premotor areas during performance of the same task, with the same apparatus, as used here. Using dynamic causal modelling (DCM), they validated a parietal-to-premotor feed-forward network node model in the context of the same experimental task that we employ here. The model is furthermore supported by other anatomical evidence of direct projections between AIP-PMv-M1 nodes in the dorsolateral circuit and pSPL-PMd-M1 nodes in the dorsomedial circuit (*Matelli et al., 1986*; *Johnson et al., 1997*; *Wise et al., 1997*; *Matelli et al., 1998*; *Luppino et al., 1999*; *Geyer et al., 2000*; *Tanne-Gariepy et al., 2002*; *Dum and Strick, 2005*; *Rushworth et al., 2006*; *Grol et al., 2007*; *Tomassini et al., 2007*; *Mars et al., 2011*; *Sallet et al., 2013*; *Neubert et al., 2014*). Our analyses are therefore based on this feed-forward model with parallel dorsolateral and dorsomedial streams.

Importantly, resting state and grasping task fMRI data sets were analysed in an identical way but were never compared directly due to categorical differences in movement artefacts which are difficult to disentangle from differences in functional interactions in the cognitive states. Because fMRI affords whole brain activity measurements, we also investigated how interactions between other nodes in the dorsolateral and dorsomedial sensorimotor circuits might dynamically reorganize in response to the paired TMS of PMv and M1.

For Experiment 2, each participant (N = 15) also underwent two fMRI scans both before and after TMS intervention. Once again the order of pre- and post-TMS scans was counterbalanced by conducting the post-TMS scan on a different day and prior to the baseline pre-TMS scan in half the subjects. The IPI was the only stimulation parameter changed in Experiment 2; an identical number of pulses were applied at the identical frequency (0.1 Hz) to identical brain areas as in Experiment 1 but now the IPI was set to 500 ms.

Data from previous fMRI studies suggest that the two areas we are investigating, PMv and M1, increase their functional connectivity during performance of the task we use (*Grol et al., 2007*). Paired-pulse TMS affords simple, direct quantification of the causal influence of one node over

another node in a given context (effective connectivity). It is known that PMv exerts an inhibitory physiological influence over M1 at rest but this turns into a facilitatory influence during grasping (*Davare et al., 2008*; *Buch et al., 2010*, *2011*). To understand how such TMS-based indices of effective connectivity related to functional connectivity indices derived from fMRI, we conducted a follow-up experiment (Experiment 3). In Experiment 3, we formally tested the relationship between effective connectivity, as measured with TMS and EMG, and fMRI-derived connectivity indices in the PMv-M1 pathway during rest and during the grasping task (N = 10).

## Experiment 1: paired stimulation of PMv and M1 at 8 ms IPI

### Changes in functional connectivity between the two stimulated areas

A qualitative sense of topographically distinct coupling patterns in the left hemisphere can be obtained by examining whole-brain co-activation maps, seeded in PMv, before and after repeated paired TMS during task performance (*Figure 2A,B*) and in the resting state (*Figure 2D,E*). In order to quantify TMS-induced changes in the relationship between left PMv and left M1, we examined the correlation between the fMRI-measured blood oxygen level dependent (BOLD) signal in the stimulated areas using the seed–based correlation analysis (SBCA) tool in the FMRIB Software Library (FSL, 56). Repeated paired TMS led to an increase in PMv-M1 BOLD functional connectivity while participants performed the task (paired t-test: t(13) = -2.59, p = 0.023; *Figure 2C*), while PMv-M1 BOLD coupling at rest was not modulated by the intervention (t(14) = -0.07, p = 0.94; *Figure 2F*).

### Changes in connectivity (psychophysiological interactions) between the two stimulated areas

Using a psychophysiological interaction (PPI) analysis (*Friston et al., 1997*; *O'Reilly et al., 2012*), we confirmed that the responsiveness of M1 to input from PMv increases following paired TMS of the

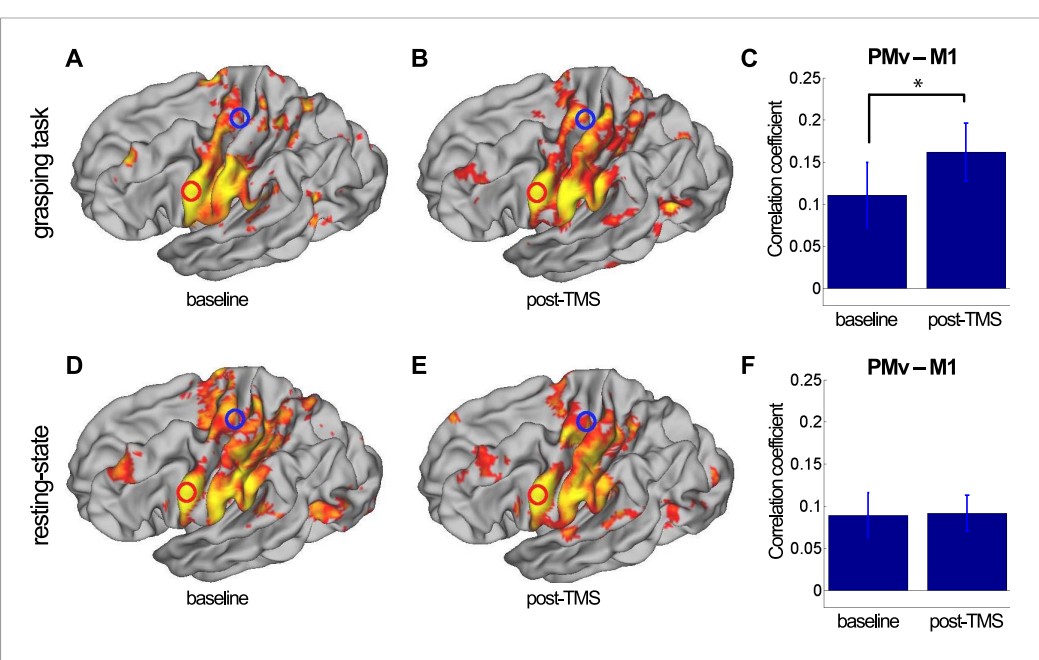

**Figure 2**. 8 ms IPI repeated paired TMS (Experiment 1). Group correlation maps seeded from PMv (red circle) during the grasping task (N = 14) (**A**, **B**) and in the resting state (N = 15) (**D**, **E**) in the baseline (**A**, **D**) and post-TMS sessions (**B**, **E**) with a spatial extent threshold of Z > 2.3 and a significance threshold of p < 0.05. There was an increase in PMv-M1 coupling during grasping (**C**), but not in the resting state (**F**). The blue circle covers the M1 ROI from which correlation coefficients were extracted. Error bars represent 1 s.e.m.

The following figure supplement is available for figure 2:

**Figure supplement 1**. 8 ms IPI repeated paired TMS (Experiment 1).

areas. Increases in connectivity between the stimulated areas (psychophysiological interactions) support the notion that PMv has a greater influence on M1 after repeated paired stimulation during performance of the prehension task (paired t-test: t(13) = −4.78, p = 0.0004; *Figure 2—figure supplement 1A*). PPI analysis of resting state fMRI data confirmed that the relationship between activity in PMv and M1 did not change in that condition following the plasticity-inducing TMS intervention (t(14) = 0.08, p = 0.93; *Figure 2—figure supplement 1B*).

### Distinct patterns of reorganization in dorsolateral and dorsomedial sensorimotor circuits

In the following analysis, we examined the broader impact of repeated paired TMS of the PMv-M1 pathway on interactions within the dorsolateral (AIP-PMv-M1) and dorsomedial (pSPL-PMd-M1) sensorimotor circuits known to be active during this grasping task (*Grol et al., 2007*). Mean Montreal Neurological Institute (MNI) coordinates of regions of interest (ROIs) are displayed in *Table 1*. We used a partial correlation analysis approach; for example, when examining pairwise PMv-AIP coupling, we did so after partialling out effects in all other nodes in the circuits: M1, PMd, pSPL, and V3A. Functional coupling measured during grasping task performance confirmed that PMv-M1 coupling was increased following repeated stimulation of those areas (PMv-M1: t(13) = -3.72, p = 0.003). At rest, repeated paired TMS led to increased dorsolateral circuit coupling (AIP-PMv: t(14) = -2.50, p = 0.025; *Figure 3*), but decreased dorsomedial circuit coupling (PMd-pSPL: t(14) = 2.22, p = 0.04; PMd–M1: t(14) = 2.84, p = 0.013; *Figure 3*). In other words, functional connectivity increased in the extended sensorimotor circuit that includes both PMv and M1 as nodes. At the same time, connectivity decreased in one of the other major sensorimotor circuits that influences M1. Finally, functional connectivity involving area V3A, a relatively early visual area that links to both the dorsomedial and dorsolateral circuits, did not change following repeated paired TMS applied to the PMv-M1 pathway (V3A-AIP: t(14) = 0.51, p = 0.618; V3A-pSPL: t(14) = 1.45, p = 0.170).

### Distinct patterns of reorganization in dorsolateral and dorsomedial sensorimotor circuits (multiple regression analysis)

Both the increase in inter-areal connectivity within the dorsolateral circuit and the parallel decrease in inter-areal connectivity within the dorsomedial circuit were corroborated with an additional analysis based on a multiple linear regression analysis with the same six sensorimotor network nodes (*Table 1*).

The analysis corroborated the finding that M1 becomes more responsive to inputs from PMv during task performance following repeated stimulation of those areas in Experiment 1 (PMv-M1: t(13) = -2.53, p = 0.0064). Again, at rest responses between more distant network node pairs were shown to alter following application of the plasticity-inducing paired-pulse TMS protocol. Within the dorsolateral circuit, PMv became more responsive to inputs from AIP (AIP-PMv: t(14) = -2.55, p = 0.023; *Figure 3—figure supplement 1A,D,E*); this finding is in line with our results from the partial correlation analysis (*Figure 3*). The multiple regression analysis also confirmed the decreased interaction of PMd and M1 within the dorsomedial circuit (PMd–M1: t(14) = 2.84, p = 0.013; *Figure 3—figure supplement 1B,D,E*); the response of PMd to activity in pSPL showed a tendency to be decreased (PMd-pSPL: t(14) = 1.78, p = 0.097; *Figure 3—figure supplement 1C,D,E*).

### Exploring reorganization in cortical networks

SBCA, partial correlation analysis, and PPI are hypothesis-driven analyses focussing on changes in functional connectivity within specific nodes of the reaching-and-grasping network. To assess the possibility that dynamic changes might occur in other neural networks and to guard against any bias in our selection of ROIs, we performed an additional exploratory analysis that employed dual-regression (*Filippini et al., 2009*). Here, voxel-wise comparisons of functional connectivity were performed across the whole brain.

First, functional networks were identified with independent components analysis (ICA) on the basis of their temporally correlated, low-frequency resting-state BOLD fluctuations. ICAs were conducted separately for fMRI data acquired during task performance and during resting state. Using the Laplace approximation for ICA dimensionality estimation, 15 and 22 large-scale spatial components—representing group-averaged neuronal networks—were extracted from the 'baseline' grasping task fMRI and from the 'baseline' resting-state fMRI of all participants, respectively. In the next stage of the analysis, two regressions were conducted: (1) the spatial regression extracted subject-specific time series for each group-averaged ICA component and (2) the

**Table 1**. Regions of interest from which BOLD time series were extracted

| ROIs in MNI standard space | x | y | z |
|---|---|---|---|
| Left M1 | −36 | −24 | 62 |
| Left PMv | −58 | 4 | 30 |
| Left PMd | −22 | −4 | 58 |
| Left AIP | −44 | −42 | 46 |
| Left pSPL | −22 | −64 | 54 |
| Left V3A | −26 | −86 | 18 |

6 mm diameter masks were created in Montreal Neurological Institute (MNI) space. Coordinates refer to MNI152_standard brain as provided by FSL. The ROI mask for M1 was based upon the meta-analysis of functional brain imaging data of motor control (*Mayka et al., 2006*). The PMv ROI location was then identified by finding the region in which BOLD activity was significantly correlated with activity in M1 at the group level, both during rest and during the grasping task. The peak resulting MNI coordinate [−58 4 30] was located in what is defined as PMv by *Mayka et al. (2006)* which had a centre-of-mass at [−52 4 24]; more specifically, it lies in the 6v/F5c subdivision of PMv identified by *Neubert et al. (2014)*. The ROI masks for AIP, PMd, pSPL, and V3A were the same size, but were centred on the group peak activation average coordinates of an fMRI study that employed a similar visually guided grasping task and the same apparatus (*Grol et al., 2007*). Masks were registered to individual EPI space in a two-step process: the mask was transformed into individual, high-resolution structural space via non-linear registration (FSL FNIRT) and then into individual functional space via affine registration (FSL FLIRT; *Jenkinson et al., 2002*).

temporal regression computed subject-specific weighted spatial maps for each group-averaged ICA component. During the last step, the weighted network masks were regressed back onto to the baseline and post-TMS fMRI time series in order to identify component networks in which BOLD correlations significantly changed following repeated paired PMv-M1 TMS.

During performance of the grasping task, a significant increase in activation of the left intraparietal sulcus area AIP [−44 −40 46] and the adjacent supramarginal gyrus (SMG) [−62 −34 34] was observed following TMS (*Figure 4A*; blue regions, p = 0.012; t-statistic images were subjected to cluster-based thresholding and corrected for multiple comparisons for a one-sided t-test at t > 1.76 with alpha-level p ≤ 0.05). This means that AIP and SMG, after paired TMS and during grasping, showed significantly more co-activation with what has previously been called the 'sensorimotor network' (*Smith et al., 2009*; *Power et al., 2011*) (red-to-yellow region, *Figure 4—figure supplement 1A*), which includes much of motor and premotor cortex and inter-connected regions of parietal cortex (*Tomassini et al., 2007*; *Mars et al., 2011*).

During rest, an analogous exploratory analysis revealed increased activity in IFJ, and adjacent areas 44d and PMv [−56 4 30] (*Figure 4B*; blue regions, p = 0.007; cluster-based thresholding and multiple comparison-correction for one-sided t-test at t > 1.76 with alpha-level p ≤ 0.05). The activity overlapped with the stimulation site of the anterior TMS coil which had been placed over the border between 6v and IFJ (*Neubert et al., 2014*) (*Figure 1*). The IFJ is a key prefrontal transitional region involved in high-level control of the motor system and interacts prominently with PMv, dorsolateral prefrontal cortex, and visual association areas in the occipitotemporal cortex (*Neubert et al., 2014*). Regions IFJ, area 44d, and PMv are part of a left-lateralised 'frontoparietal network' (red-to-yellow region, *Figure 4—figure supplement 1B*) and the results mean that they become more coupled with the rest of this network following repeated paired-pulse TMS.

## Experiment 2: paired stimulation of PMv and M1 at 500 ms IPI

Since increased functional connectivity between two areas is difficult to distinguish from increased mean firing in the two areas, it is possible that the measured connectivity changes observed following repeated paired PMv-M1 TMS at 8 ms IPI in Experiment 1 could have been the result of an increase in activity in each of the stimulated areas instead of a induced change in functional connectivity (*Chawla et al., 2000*). In Experiment 2, we therefore applied identical numbers of pulses at identical frequencies over the identical brain regions, but we did so at 500 ms IPI. This interval is many times longer than the longest one at which PMv-M1 interactions have been observed (*Davare et al., 2008*; *Neubert et al., 2010*). While such a protocol ought to induce similar changes in each stimulated region, it should not result in their co-activation or in STDP. Using a higher-level analysis (mixed-model ANOVA) with between-subjects factor 'PROTOCOL', we directly contrasted the effects from Experiment 1 and Experiment 2 for each of the analyses conducted. We present the results in *Table 2* and will go through the findings in the following order: (1) functional connectivity analysis

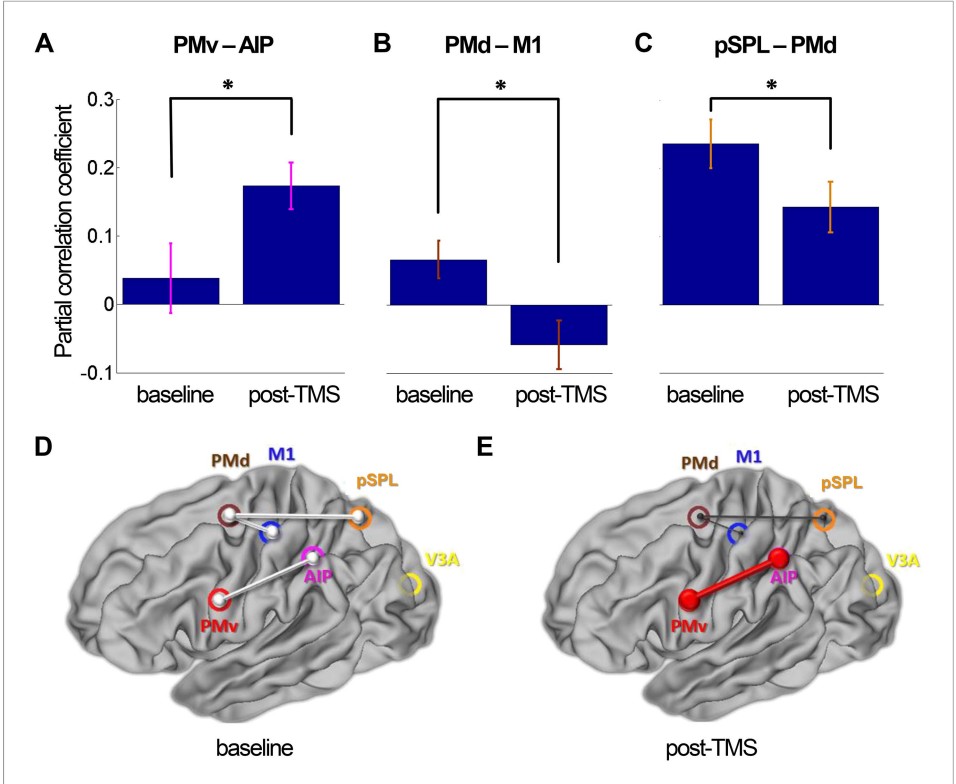

**Figure 3**. 8 ms IPI repeated paired TMS (Experiment 1). Partial correlation analysis of resting-state fMRI. There was a significant increase in coupling between other nodes (PMv-AIP) within the dorsolateral sensorimotor network that are linked to PMv and M1 over which repeated paired stimulation was applied (**A**). At the same time, there were significant decreases in functional connectivity within the dorsomedial sensorimotor network; between PMd and M1 (**B**) and PMd and pSPL (**C**). Error bars represent 1 s.e.m. (**D** and **E**) Schematic representation of mean group connectivity weights (grey lines) in baseline and post-TMS sessions. (**D**) All weights are standardised to the baseline partial connectivity of each connection. (**E**) Significant increments in PMv-AIP connectivity (red line) and decrements in PMd-M1 and PMd-pSPL connectivity (black lines) in the post-TMS session (N = 15).

The following figure supplement is available for figure 3:

**Figure supplement 1**. 8 ms IPI repeated paired TMS (Experiment 1).

between PMv-M1; (2) PPI analysis between PMv-M1; (3) partial correlation analysis between pairwise network nodes; (4) multiple regression PPI analysis between pairwise network nodes; and (5) dual-regression analysis.

A higher-level analysis of functional connectivity between PMv-M1 confirmed that PMv-M1 coupling was not changed during task performance following paired TMS with a 500 ms IPI; this is in contrast to significantly greater connectivity following paired TMS with an 8 ms IPI (mixed-model ANOVA: TIME by PROTOCOL interaction: $F_{(1,26)} = 4.64$, $p = 0.041$; Experiment 2 during task: paired t-test: $t(13) = 0.94$, $p = 0.36$). At rest, functional connectivity was not changed in the PMv-M1 connection following either protocol (Experiment 2 at rest: paired t-test: $t(14) = 0.07$, $p = 0.95$). A higher-level PPI analysis of PMv-M1 connectivity supports the finding from the functional connectivity analysis (mixed-model ANOVA: TIME by PROTOCOL interaction during task: $F_{(1,26)} = 6.92$, $p = 0.014$; Experiment 2 during task: paired t-test: $t(13) = 0.98$, $p = 0.35$). At rest, no changes in PMv-M1 connectivity were found either (Experiment 2 at rest: paired t-test: $t(14) = 0.20$, $p = 0.85$).

In parallel to the analyses for Experiment 1, we then examined the wider dorsolateral and dorsomedial sensorimotor circuits. A partial correlation analysis contrasting Experiment 1 with Experiment 2 confirmed that during task, PMv-M1 coupling was only changed in the grasping

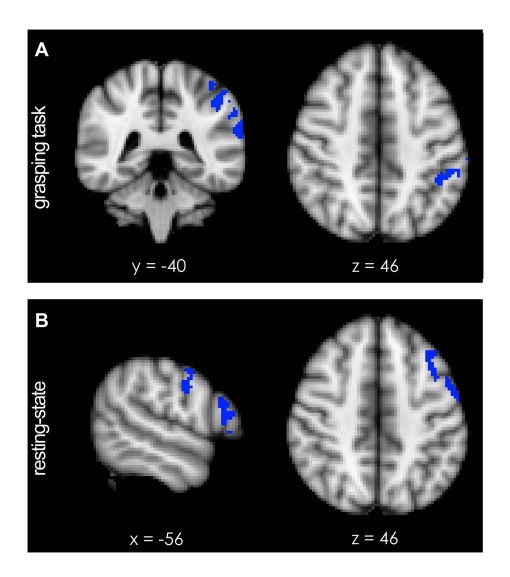

**Figure 4**. Short-term potentiation of PMv-M1 connectivity led to increased activation of left AIP [−44 −40 46] (blue indicates areas of increased coupling) during grasping. (**A**) AIP activity became significantly more coupled with the sensorimotor network (see **Figure 4—figure supplement 1A**). (**B**) At rest, pathway potentiation evoked coactivation of PMv [−56 −4 30] and a prefrontal region close to the site of stimulation (inferior frontal junction (IFJ) [−56 20 22]). These areas (blue) became specifically more coupled with a left-lateralised frontoparietal network (see **Figure 4—figure supplement 1B**). All effects (p < 0.05; N = 15).

The following figure supplement is available for figure 4:

**Figure supplement 1**. Functional spatial networks (red-to-yellow, z-statistical map thresholded at z = 5.0) as defined by synchronous fluctuations in BOLD activity during performance of the prehension task (**A**) and during rest (**B**).

condition following plasticity induction with an 8 ms IPI (mixed-model ANOVA: TIME by PRO-TOCOL interaction: $F(1,26) = 7.47$, $p = 0.011$; Experiment 2 during task: paired t-test: $t(13) = 1.18$, $p = 0.26$).

At rest, no changes in coupling in the wider dorsolateral and dorsomedial sensorimotor circuits were found after 500 ms IPI TMS (Experiment 2 at rest: paired t-tests: AIP-PMv: $t(14) = 1.08$, $p = 0.30$; pSPL-PMd: $t(14) = -1.24$, $p = 0.24$; PMd-M1: $t(14) = 0.47$, $p = 0.65$). Further statistical testing showed that the strengthening of functional connectivity in the AIP-PMv pathway was significantly stronger after 8 ms IPI TMS (Experiment 1) than was after 500 ms IPI TMS (Experiment 2) (mixed-model ANOVA: TIME by PROTOCOL interaction: $F(1,28) = 7.15$, $p = 0.012$) as were the decreases in the PMd-pSPL pathway strength (mixed-model ANOVA: TIME by PROTOCOL interaction: $F(1,28) = 5.29$, $p = 0.029$). The PMd-M1 connection showed a similar trend (mixed-model ANOVA: TIME by PROTO-COL interaction: $F(1,28) = 5.92$, $p = 0.08$).

The lack of reorganisation within the dorso-lateral and dorsomedial circuits in Experiment 2 was confirmed by employing the same multiple regression PPI analysis used for Experiment 1. **Table 2**.

Finally, employing a dual-regression analysis, we confirmed that it was only after 8 ms IPI TMS in Experiment 1 that the left frontoparietal network was found to be more coherently coupled with itself and co-active with PMv in the resting state, but not after 500 ms IPI TMS in Experiment 2 (mixed-model ANOVA: TIME by PROTOCOL interaction: $p = 0.022$). The fronto-parietal network did not significantly alter its coupling pattern in Experiment 2 ($p = 0.446$).

## Experiment 3: comparison of connectivity measures: paired-pulse TMS-derived effective connectivity contrasted with fMRI-derived functional connectivity

In Experiment 3, we investigated how both TMS-based effective connectivity and fMRI-based functional connectivity indices relate to each other within the same subjects, focussing on the PMv-M1 pathway in ten of the subjects tested in Experiment 1. We measured the size of MEPs evoked by TMS of M1 alone and evoked by M1 TMS applied 8 ms after a PMv pulse. Such PMv TMS pulses are known to either augment or diminish the size of the MEP induced by M1 TMS depending on whether or not subjects are making grasping movements or are at rest, respectively (*Davare et al., 2008*; *Buch et al., 2010*, *2011*). To quantify the influence of PMv over M1, we compared the MEPs induced by M1 stimulation alone with MEPs induced by M1 stimulation that was preceded by PMv-stimulation. A TMS-based index of effective connectivity between PMv and M1 was calculated as the ratio of the difference in MEP amplitudes evoked by paired-pulse TMS and single-pulse TMS divided by single-pulse-evoked MEP amplitudes. The PMv-M1 TMS ratio is positive when PMv TMS augments the size of M1 TMS-induced MEPs, but negative when PMv TMS diminishes M1 TMS MEP size. Moreover, the ratio was measured both while subjects were at rest and during the reaching task. In this way, the

**Table 2**. Summary of results from hypothesis-driven analyses conducted on 8 ms-IPI Experiment 1 and control Experiment 2 (IPI of 500 ms)

| | | Expt 1 (IPI 8 ms) | | | | Expt 2 (IPI 500 ms) | | | | Expt 1 vs Expt 2 | | | |
|---|---|---|---|---|---|---|---|---|---|---|---|---|---|
| | | PMv-M1 | AIP-PMv | pSPL-PMd | PMd-M1 | PMv-M1 | AIP-PMv | pSPL-PMd | PMd-M1 | PMv-M1 | AIP-PMv | pSPL-PMd | PMd-M1 |
| Functional connectivity (fc) | grasp | t(13) = −2.59; p = 0.023* | | | | t(13) = 0.94; p = 0.36 | | | | F(1,26) = 4.64; p = 0.041* | | | |
| | rest | t(14) = −0.07; p = 0.94 | | | | t(14) = 0.07; p = 0.95 | | | | n.a. | | | |
| partial correlation fc | grasp | t(13) = −3.72; p = 0.003* | n.s. | n.s. | n.s. | t(13) = 1.00; p = 0.34 | n.s. | n.s. | n.s. | F(1,26) = 7.76; p = 0.011* | n.s. | n.s. | n.s. |
| | rest | t(14) = −0.07; p = 0.95 | t(14) = −2.50; p = 0.025* | t(14) = 2.22; p = 0.04* | t(14) = 2.84; p = 0.013* | t(14) = −0.39; p = 0.70 | t(14) = 1.08; p = 0.30 | t(14) = −1.24; p = 0.24 | t(14) = 0.47; p = 0.65 | n.a. | F(1,28) = 7.15; p = 0.012* | F(1,28) = 5.29; p = 0.029* | F(1,28) = 5.92; p = 0.08 |
| Psycho-physiological interaction (PPI) | grasp | t(13) = −4.78; p = 0.0004* | | | | t(13) = 0.98; p = 0.35 | | | | F(1,26) = 6.92; p = 0.014* | | | |
| | rest | t(14) = 0.08; p = 0.93 | | | | t(14) = 0.20; p = 0.85 | | | | n.a. | | | |
| multiple regression PPI | grasp | t(13) = −2.53; p = 0.0064* | n.s. | n.s. | n.s. | t(13) = 1.18; p = 0.26 | n.s. | n.s. | n.s. | F(1,26) = 7.47; p = 0.011* | n.s. | n.s. | n.s. |
| | rest | n.a. | t(14) = −2.55; p = 0.023* | t(14) = 1.78; p = 0.097 | t(14) = 2.84; p = 0.013* | n.a. | t(14) = 0.41; p = 0.96 | t(14) = −1.18; p = 0.26 | t(14) = 0.01; p = 0.99 | n.a. | F(1,28) = 5.74; p = 0.024* | F(1,28) = 3.66; p = 0.066 | F(1,28) = 4.44; p = 0.044* |

Analyses were conducted on rest and task data. Moreover in order to show that specific effects relate to plasticity induction (8 ms IPI) several higher-level analyses contrasting Experiment 1 and 2 are presented. T-tests were conducted as two-tailed paired t-tests (within subjects). Mixed-model ANOVAs were conducted between experiments (across subjects). Detailed information on all analyses is provided in the 'Materials and methods' section. Asterisks indicate significant results, p < 0.05. Abbreviations: n.s. = non-significant.

influence of PMv over M1 could be quantified for both cognitive states, and the direction (facilitatory or inhibitory) and magnitude of effective connectivity could then be compared to functional connectivity as measured with fMRI in Experiment 1.

During task performance, the two connectivity measures (a TMS-based effective connectivity index and the fMRI-based functional connectivity measure) for the PMv-M1 pathway were positively correlated across subjects at baseline (Pearson's correlation coefficient: R = 0.74, p = 0.01) (*Figure 5A*). A positive correlation indicates that the greater the facilitatory influence of PMv on M1 as measured with TMS, the greater the fMRI-derived functional connectivity. Furthermore, during task performance, paired-pulse TMS-derived effective connectivity was still significantly correlated with fMRI-derived connectivity after repeated paired 8 ms IPI TMS (R = 0.87, p = 0.0008) (*Figure 5B*).

At rest, the correlation between the two measures was not significant at baseline (R = −0.21, p = 0.57) (*Figure 5C*). However, a significant correlation was observed between the baseline TMS-derived effective connectivity measure and the fMRI-derived functional connectivity measure following repeated paired 8 ms IPI TMS (R = −0.68, p = 0.03) (*Figure 5D*). Intriguingly, the correlation was negative which implies that when PMv had a stronger net inhibitory influence on M1 (as indexed by neurophysiological measurements), fMRI indicated stronger net positive functional connectivity between the two areas across participants. Finally, we note that baseline effective connectivity strength at rest (i.e., inhibitory) and during grasping (i.e., excitatory) were also negatively correlated across individuals (R = −0.63, p = 0.049).

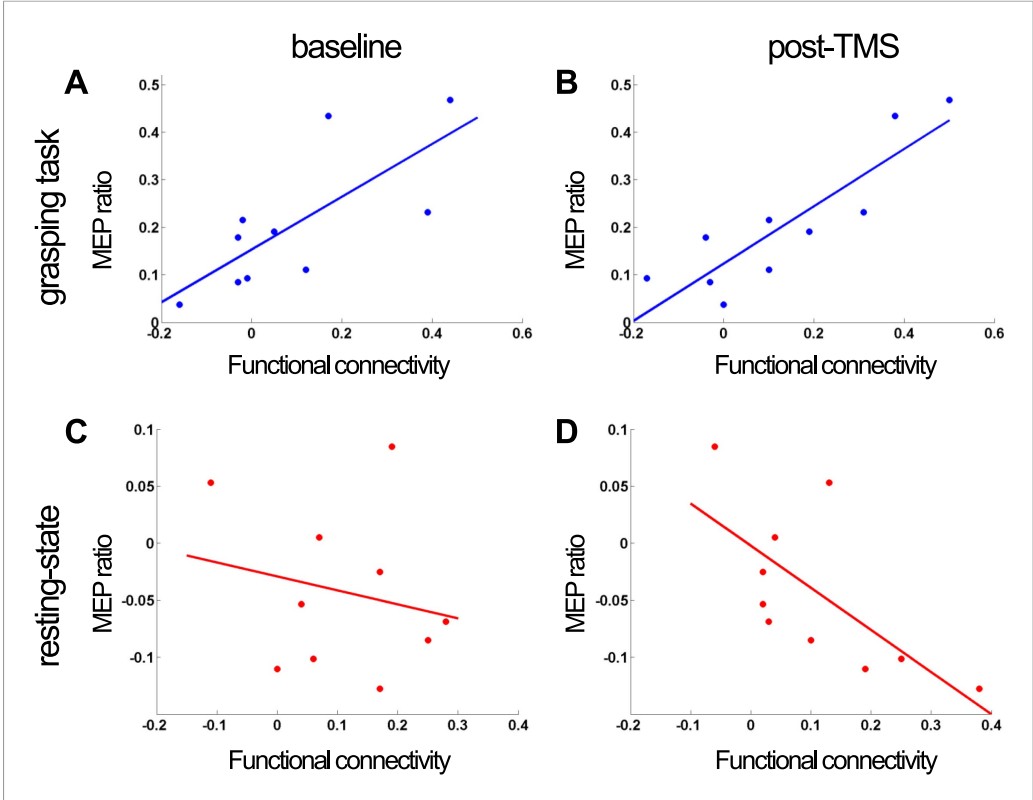

**Figure 5**. Experiment 3: correlation of PMv-M1 connectivity measures before and after 8 ms IPI paired TMS (N = 10). When subjects were making grasping movements, there was a significant correlation between functional connectivity (derived from partial-correlation analysis of fMRI) in the baseline (**A**) and post-TMS session (**B**) and the baseline effective connectivity measure derived from the paired pulse TMS MEP ratio at baseline. There was a significant negative correlation between functional connectivity in the post-TMS session and the baseline effective connectivity measure derived from the paired-pulse TMS MEP ratio at baseline (**D**). The correlation did not reach significance when the functional connectivity measure as well as the effective connectivity measure was taken from the baseline session (**C**).

## Discussion

In this study, we describe the functional connectivity signature in fMRI data of short-term synaptic potentiation within a specific anatomical pathway. Using two different paired-pulse TMS manipulations, we demonstrated that application of a TMS protocol known to change synaptic efficacy within a motor pathway (PMv-M1) results in increases in functional connectivity along the same pathway that can be measured with fMRI. Furthermore, we established a significant correlation between the size of the causal influence of PMv on M1 (effective connectivity as assessed by paired-pulse TMS) and the fMRI-based index of functional connectivity across individuals.

PMv provides one of the principal inputs into M1, and it exerts a powerful influence over M1 output (*Shimazu et al., 2004*; *Dum and Strick, 2005*) but the degree of interaction between PMv and M1 can be modulated by repeated paired pulse TMS with an 8 ms IPI over PMv and M1 (*Buch et al., 2011*). By means of two correlation analyses and two PPI analyses on fMRI data acquired during the performance of a grasping task, we confirmed our a priori hypothesis that augmented pathway efficacy is mirrored in increases in inter-regional functional connectivity. We specifically show that the activity in PMv was more tightly related to activity in M1 following intervention (*Figure 2C*) and that the influence of PMv on M1 (or the responsiveness of M1 to input from PMv) was increased in response to short-term pathway manipulation (*Figure 2C* and *Figure 2—figure supplement 1A*). In the current study, we did not track the duration of these changes in functional coupling after the

intervention. However, we note that in a previous study changes in effective connectivity were shown to last more than 1 hr (*Buch et al., 2011*).

An alternative interpretation of the increased correlation in PMv and M1 activity might be attempted not by referring to synaptic change involving the pathway between them but by simply referring to the changes that pulses over each area induce even when applied in isolation. Such an explanation, however, is unlikely to be correct. First, there is no empirical reason to think that TMS stimulation of any one area at a frequency of 0.1 Hz would lead to a protracted change in that area's activity which is detectable by fMRI many minutes later. Moreover, Experiment 2 employed a control procedure, a repeated paired-pulse TMS protocol which precludes the temporal contiguity required for pathway plasticity induction by using a longer IPI (500 ms vs 8 ms in Experiment 1) even though it involved stimulation of the same areas at the same frequency and intensity. Functional connectivity between PMv and M1 was not altered in response to the control procedure in Experiment 2 suggesting that the influence of PMv on M1 in Experiment 1 was indeed attributable to short-term changes in synaptic efficacy.

These observations extend previous studies that described acute compensatory plasticity of the motor system following single-site TMS manipulations albeit in the context of task performance (*Lee et al., 2003*; *O'Shea et al., 2007*; *Grefkes et al., 2010*; *Hartwigsen et al., 2012*). The results also extend the understanding of the effect of the repeated paired pulse TMS plasticity induction procedure that we previously examined in the absence of fMRI data (*Buch et al., 2011*). For example, greater M1 output was previously observed by measuring MEP sizes with M1 TMS during grasping after the paired pulse plasticity induction procedure but the origin of the effect was unclear. The new results make it clear that it is driven by M1 being more responsive to activity in PMv.

Moreover, the current results reveal that the increase in PMv-M1 connectivity was very specific. Although several analyses demonstrated that functional connectivity also increased between PMv and its principal parietal input in the dorsolateral sensorimotor pathway, AIP and adjacent parts of the parietal cortex (*Godschalk et al., 1984*; *Davare et al., 2010*) (*Figures 3A,D,E,4B*, *Figure 4—figure supplement 1B*), increased functional connectivity was not seen between M1 and other premotor areas. In fact the reverse was true; functional connectivity between PMd and M1 significantly decreased. In addition, functional connectivity in other parts of the dorsomedial sensorimotor circuit, between PMd and pSPL, also declined (*Figure 3D,E*).

The exact functional role of these accessory decreases in functional coupling in distant connections requires further investigation. It is unclear whether they should be thought of as 'compensatory' as they were more prominent at rest than during the grasping task. Inhibitory plasticity might accompany excitatory plasticity in order to stabilise neural networks involved in learning (*Vogels et al., 2011*). They suggest that inhibitory spike timing-dependent plasticity could balance excitatory inputs. Learning or the formation of associative excitatory connections in such networks would require the co-adaptation of excitatory and inhibitory synapses. Although Vogels' et al. ideas largely make predictions about structural and functional properties of local neural circuits, the results of this experiment could be taken to suggest that similar principles apply to the network and systems level.

Additionally enhancement of one pathway might be accompanied by diminution of a parallel pathway if both of them compete to influence a particular target structure such as M1. It has been argued that two pathways for movement preparation—the dorsomedial visuomotor stream (pSPL–PMd) and the dorsolateral stream (AIP–PMv)—complement each other by driving movement selection proportional to the amount of information available in each stream (*Verhagen et al., 2008*). It remains to be determined how exactly movement selection is biased towards dorsolateral or dorsomedial streams and whether there are categorical or graded differences. The study of multi-sensory integration has generated proposals concerning how integration of information from two different streams might be achieved (*Ernst and Banks, 2002*).

Future research needs to understand the relation of these different pathways and how they interact and potentially compete to guide movement selection. With more detailed knowledge about the structural skeleton and the functional relationship of these streams, we might be able to predict the complex effects of learning and plasticity not only on the particular network primarily involved in learning and plasticity but also on other parallel streams and networks. More generally this might eventually contribute to a better understanding of network effects relating to learning, development and degeneration (*Fair et al., 2008*; *Seeley et al., 2009*; *Dayan and Cohen, 2011*). For this line of research whole-brain approaches such as fMRI or magnetoencephalography (MEG) might have some

advantages in some contexts in comparison to examining more local effects of plasticity, such as changes in MEPs (*Buch et al., 2011*).

The absence of reduced dorsomedial pathway coupling in Experiment 2 suggests that it cannot simply be due to the repeated asynchrony of activity in PMd and M1 that is induced by the paired TMS protocol (repeated TMS-induced activation of M1 without corresponding activation of PMd). If this were the case, then one would expect the protocol in Experiment 2, in which PMd and M1 activity was also stimulated asynchronously, to induce similar decrements in dorsomedial pathway coupling, but this was not observed in the current study.

Some of the changes in functional connectivity seen in Experiment 1 were more apparent when subjects were engaged in the motor task while others were more apparent when subjects were at rest. Broadly speaking, changes in the interactions between the stimulated areas themselves, PMv and M1, areas known to be intimately involved in the making of grasping movements, were most apparent when subjects were actively engaged in just such motor activity (*Figure 2C*, *Figure 4A*, *Figure 2—figure supplement 1A*, *Figure 4—figure supplement 1A*). By contrast, changes in interactions between PMv and adjacent ventral frontal areas such as IFJ with other prefrontal and parietal areas concerned with high-level cognitive control and attention were more apparent when subjects were at rest (*Figure 4B*). Changes in interactions between PMv and the parietal areas surrounding AIP were apparent both when subjects were engaged in grasping and when they were at rest (*Figure 4A,B*). The differential sensitivity of the two conditions to different aspects of functional connectivity change may be related to the varying roles of these pathways in the control, planning, and coordination of movement and the actual implementation and execution of movements.

Finally, Experiment 3 demonstrated that similar patterns of fMRI-measured functional connectivity are associated with either net facilitatory or net inhibitory influences being exerted by PMv over M1 when subjects are engaged in a grasping task or at rest (*Figure 5*). Despite these differences in the sign of the relationship between separate indices of connectivity, the sizes of functional connectivity indices were correlated across subjects, that is, subjects with stronger functional connectivity between PMv and M1 also showed higher degrees of TMS-measured effective connectivity between the two areas. The paired pulse TMS approach provides a particularly direct and simple assay of the effective connectivity that exists between two brain areas but it complements other techniques, such as DCM, that attempt to recover effective connectivity estimates from fMRI data in particular behavioural contexts (*Friston, 1994*; *Friston et al., 2003*).

From our experiments, we infer that functional connectivity is not only shaped by structural connections but also by short-term plastic changes in synaptic efficacy. It still, however, remains a challenge to link the changes seen with neuroimaging measures to specific cellular and molecular level changes at the synapse. Paired stimulation of two brain regions led to increased functional connectivity between the two regions but also to a limited set of other functional connectivity changes, both positive and negative, in other parts of the cortical sensorimotor circuits. In addition, we showed that positive functional connectivity between two areas may reflect either facilitatory or inhibitory effective connectivity. Such changes in functional connectivity are not only interesting in their own right but also because different patterns of premotor–M1 interaction are seen in patients who do and who do not recover motor skills after stroke (*Gerloff et al., 2006*; *Lotze et al., 2006*). An interesting possible future avenue for research is to employ pathway-specific non-invasive stimulation protocols in patients to induce directed changes in connectivity and thereby potentially drive neural network reorganisation so as to assist in recovery of motor function.

## Materials and methods

### Volunteers

15 subjects (eight males) participated in Experiment 1; fifteen subjects (nine males) participated in Experiment 2. For Experiment 3, paired-pulse TMS data were obtained for 10 participants from Experiment 1 (five males). The overall mean age of all participants was 24 ± 4 years (mean ± SD). The study was approved by the local ethics committee and informed consent was obtained from all subjects.

## Transcranial magnetic stimulation (TMS)

TMS was applied using two Magstim 200 stimulators each of which was connected to a 50 mm figure-8 coil. On a day prior to the day of the combined TMS-fMRI experiment, resting motor threshold (RMT) was determined for each participant for the left M1 'hot spot', which is the scalp location where TMS evoked the largest MEP amplitude in right first dorsal interosseous (FDI) (*Rossini et al., 1994*) (mean ± SD: 40 ± 7% stimulator output). Electromyographic (EMG) activity in right FDI was recorded with bipolar surface Ag-AgCl electrode montages. Responses were bandpass filtered between 10 and 1000 Hz, with additional 50 Hz notch filtering, sampled at 5000 Hz, and recorded using a CED 1902 amplifier, a CEDmicro1401 Mk.II A/D converter, and PC running Spike2 (Cambridge Electronic Design).

To stimulate left M1, one coil was placed over the scalp location of the left FDI 'hot spot' at average MNI coordinates [−40 −18 59]. The location was projected onto the high-resolution, T1-weighted MRI brain scan of each participant using frameless stereotactic neuronavigation (Brainsight; Rogue Research). The second coil, over left PMv, was positioned so as to be ventral to the convergence of the inferior frontal sulcus and inferior precentral sulcus on each individual's MRI scan. The mean MNI location [−56 19 19] was within the region defined previously as human PMv (*Mayka et al., 2006*) but which more precisely corresponds to the border between IFJ and 6v (*Neubert et al., 2014*) (*Figure 1B*). As in previous studies PMv was stimulated with 110% of RMT and M1 with a stimulation intensity sufficient to elicit a 1-mV MEP following a single TMS pulse (*Neubert et al., 2010*; *Buch et al., 2010 and 2011*). For the duration of the experiment, TMS coils were fixed in place tangentially to the skull by means of adjustable metal arms and monitored throughout the experiment. In all three experiments, an attempt was made to induce plasticity between PMv and M1 by repeated paired stimulation of the two areas. Paired TMS lasted for 15 min and was applied at a frequency of 0.1 Hz (i.e., 90 pairs of pulses), with an IPI of 8 ms (Experiments 1 and 3) and an IPI of 500 ms (Experiment 2).

In Experiment 3, 8 ms IPI paired pulses were also used in a second way in order to provide a neurophysiological index of effective connectivity between PMv and M1. Ten participants drawn from Experiment 1 took part on a day separate from the two MRI image acquisition days. MEPs were recorded from the right FDI muscle in response to either M1 TMS (20 trials) or paired-pulse TMS delivered over both PMv and, 8 ms later, over M1 (20 trials). Trials were administered in pseudorandom order. The ratio of MEP sizes in the paired pulse trials compared to the single M1 pulse trials provided an index of the modulatory influence of PMv over M1.

Experiment 3 was conducted under two conditions. In the grasping condition, volunteers sat in a darkened room and made right-hand reaching and grasping movements cued by illumination of one of two concentrically arranged cylinders (15 and 65 mm diameter) located 30 cm in front of the starting hand position (*Buch et al., 2010 and 2011*). Each trial was initiated by pressing a touch bar with the right hand. Intertrial intervals were therefore variable (mean ± SD, 6.90 ± 0.79 s) but did not differ significantly across phases of each experiment. Following a variable delay of 5–7 s (uniformly distributed), one cylinder was illuminated. Volunteers responded by grasping it with their thumb and index finger before lifting it from its pedestal. Reaction and movement times were recorded. All trials were accompanied by either M1 TMS or paired PMv-M1 TMS, with the pulse applied to M1 always occurring 100 ms after cylinder illumination, which was before movement onset.

For TMS when at rest, volunteers still attended to cylinder illumination, as they had done during the motor task, but now they simply maintained a static hand posture. To control for the overall temporal distribution of the TMS pulses, ITIs for rest blocks were defined as the sum of the ITIs used in task blocks plus a reaction and movement time sample drawn from probability density functions for these variables (*Buch et al., 2010 and 2011*). ITIs were therefore variable (mean ± SD, 6.23 ± 0.07 s) but did not differ significantly across phases of each experiment.

## FMRI acquisition

MRI data were acquired on a Siemens 3T Trio MRI scanner at the Oxford Centre for Clinical Magnetic Resonance Imaging (OCMR). For purposes of neuronavigation-guided TMS, all volunteers underwent high-resolution, T1-weighted structural MRI scans that included nose and ears. For each condition—resting state and grasping task—5 min of whole-brain T2*-weighted gradient echo planar images (EPIs) sensitive to BOLD were acquired (repetition time = 3.000 ms, echo time = 30 ms, flip angle = 87°, isotropic voxels of 3.0 mm, no slice gap, 45 slices in axial direction).

Participants were instructed to keep their eyes closed during resting-state fMRI. During the grasping task, which was based on a previous study (*Grol et al., 2007*; *Majdandžić et al., 2007*), participants performed 66 reaching-and-grasping trials towards either a small or a large cube positioned in front of them. A new trial sequence was generated for every participant and for each session, with an inter-trial interval of 4295.5 ms–4795.5 ms (mean ± SD: 4545.5 ms ± 145.5 ms) which allowed every participant to complete the movement. Participants lay supine in the MR scanner with the eight-channel head coil tilted forward by 30° enabling them to perform a naturalistic visually guided reaching-and-grasping task in front of their bodies (*Figure 1C*). Participants were allowed to move their eyes in order to guide their movements. An optical response button box was placed on their right upper leg and served as a start-and-finish position. Reaction times and total movement times were recorded. With the aim of avoiding movement artefacts, the participant's upper arm lay on a wedge-shaped polyfoam cushion and was firmly, but comfortably strapped to the side of the participant's chest. This setup constrained rotation movements in the plane between the button box and the target objects. The head was supported with foam wedges. The participants had received extensive training in the reaching-and-grasping task at least one day prior to the first MRI acquisition outside the MRI scanner. The target object, which consisted of a large red cube and a small green cube (*Figure 1C*, inset), was held in place through an arc-shaped device positioned over each participant's hips. Participants had been instructed to grasp one of the two cubes, to slide it out of its supporting rail on a rectangular box, and to return it into the same supporting rail. On a given trial, either the large red or the small green cube was to be grasped. A red or green light-emitting diode (LED) in the middle of the rectangular box instructed the participant which cube to grasp. MRI-compatible switches on the device recorded the time at which the object was removed from the supporting rail and the time at which the object was returned into the supporting rail. Control of LEDs and recording of movement-related responses was performed with a computer running Presentation 15.0 (Neurobehavioral Systems, San Francisco, CA). TMS was applied outside the MRI scanner room. Participants walked to the MRI scanner and scanning commenced within 3 to 4 min. Note, previous neurophysiological experiments (*Buch et al., 2011*) suggest plasticity induction should last at least 1 hr with this protocol and that there were no differences in efficacy immediately after intervention in comparison to +30 min or +60 min post-intervention.

## Image pre-processing

FMRI data were pre-processed using tools from the FMRIB Software Library (FSL; www.fmrib.ox.ac.uk/fsl; *Smith et al., 2004*). Imaging volumes were registered to the individuals' structural scan using boundary-based registration (BBR) (*Greve and Fischl, 2009*) and to standard space using FMRIBs Linear Image Registration Tool (FLIRT) with 12° of freedom. Pre-processing involved: motion correction (McFLIRT), brain extraction (BET), spatial smoothing with a Gaussian 5 mm full-width at half-maximum (FWHM) kernel, and high-pass temporal filtering at 100 s.

## Image pre-processing for dual-regression

Individual subject independent-component analysis (ICA) fMRI analysis was carried out on baseline data of twelve Experiment 1 and eleven Experiment 2 data sets using Multivariate Exploratory Linear Optimized Decomposition into Independent Components (MELODIC) (*Beckmann and Smith, 2005*). Individual pre-statistical processing consisted of motion correction (McFLIRT), brain extraction (BET), spatial smoothing using a Gaussian kernel of full-width at half maximum (FWHM) of 5 mm, and high-pass temporal filtering. Imaging volumes were registered to the individuals' structural scan using boundary-based registration (BBR) (*Greve and Fischl, 2009*) and to standard space using FMRIBs Linear Image Registration Tool (FLIRT) with 12° of freedom. Pre-processed functional data were temporally concatenated across subjects.

## Seed-based correlation analysis (SBCA)

SBCA maps the functional connectivity of one 'seed' ROI across the entire brain in a voxel-wise manner on the basis of the correlation between the seed ROI's BOLD time series and the BOLD time series at each voxel in the rest of the brain (*O'Reilly et al., 2010*). We employed SBCA to assess if paTMS-based modulation of the PMv-M1 pathway dynamically altered the functional interactions of either of these two nodes with each other and/or with other nodes within the reaching and grasping

network. We assessed the functional connectivity of a 6 mm diameter seed mask in left PMv with the whole brain (target mask) before and immediately after paTMS and contrasted PMv-M1 connectivity at baseline vs connectivity during post-TMS (for details about statistical analyses see below). The analyses of resting state and grasping task fMRI data were conducted independently. All analyses conducted for Experiment 1 and Experiment 2 were identical, which allowed us to directly contrast the effects in a higher-level analysis. For the first step of SBCA, statistical connectivity maps for every individual and for each of the four conditions (resting-state baseline/resting-state plasticity expression and task baseline/task plasticity expression) were created using the SBCA tool implemented in FSL (fsl sbca).

The time series for the left PMv seed mask was calculated. The SBCA model also accounted for the time series resulting from structured noise in the average BOLD signal in white matter (WM), grey matter (GM), and cerebrospinal fluid (CSF) and head movement (six regressors resulting from McFLIRT motion correction). WM, GM, and CSF masks were derived from individual T1-weighted structural images using the FSL segmentation tool FAST and registered to EPI space using FLIRT (*Jenkinson and Smith, 2001*; *Zhang et al., 2001*). The resulting connectivity map described the correlation between the average BOLD time series of the PMv mask and the time series for each voxel within the whole brain. The individual correlation maps were transformed into MNI space, using FLIRT affine registration of EPI to structural space and subsequently FNIRT non-linear registration to MNI space. Standard space group correlation maps (z-score maps) were generated by entering SBCA-derived individual correlation maps into a group general linear model (GLM) and thresholding at Z > 2.3 with a significance threshold of p < 0.05. These thresholded group z-score maps were projected onto the Midthickness.32k CaretBrain as provided by the Human Connectome Project Workbench using the 'surf proj' algorithm as implemented in FSL and then visualized using the Human Connectome Project Workbench (http://www.humanconnectome.org/connectome/get-connectome-workbench.html).

As the next step, we computed the average time series resulting from these statistical connectivity maps for the M1 ROI. We then compared time series correlations of PMv with M1 at baseline and during post-TMS. For statistical comparisons, we conducted a paired t-test, contrasting PMv-M1 interactions at baseline vs during post-TMS. Prior to statistical analysis, correlation coefficients were Fisher z-transformed. We also conducted a higher-level analysis, contrasting Experiment 1 with Experiment 2, using a mixed-model ANOVA with within-subject factors TIME (baseline / post-TMS) and between-subjects factor PROTOCOL (Experiment 1/Experiment 2). We used a significance level of p < 0.05. 15 participants contributed to the resting-state group z-score map (for both Experiment 1 and Experiment 2); 14 participants contributed to the grasping task group z-score map, since the data of one participant had to be removed due to excessive head movement during data acquisition (for both Experiment 1 and Experiment 2).

## Partial correlation analysis

To investigate changes in functional connectivity between pairs of grasping network nodes, we conducted a partial correlation analysis between the BOLD time series of directly connected nodes of the left hemisphere using Matlab R2013b (MathWorks). Partial correlation analysis generated correlations represent only correlations specific to the pair of cortical regions in question by regressing out the time series of all other network nodes under investigation. We focussed on pairs of regions thought to be monosynaptically connected (*Matelli et al., 1986*; *Johnson et al., 1997*; *Wise et al., 1997*; *Matelli et al., 1998*; *Luppino et al., 1999*; *Geyer et al., 2000*; *Tanne-Gariepy et al., 2002*; *Dum and Strick, 2005*; *Rushworth et al., 2006*; *Grol et al., 2007*; *Tomassini et al., 2007*; *Mars et al., 2011*; *Sallet et al., 2013*; *Neubert et al., 2014*): M1-PMv, PMv-AIP, AIP-V3A, M1-PMd, PMd-pSPL, and pSPL-V3A. Individual BOLD time series for each network node mask (6 mm diameter) were generated using a GLM-based design that incorporated regressors denoting potentially confounding factors such as variation in WM, GM, and CSF, and whole brain BOLD signal as implemented in FSL (fsl glm). Individual partial correlations were normalised using Fisher's z-transform.

Analogous to statistical tests used in SBCA, we conducted paired t-tests contrasting pairwise interactions at baseline vs during post-TMS on Fisher z-transformed partial correlation coefficients (independently for Experiment 1 and Experiment 2). At a later stage, we also subjected Experiment 1

and Experiment 2 to a direct comparison by means of a mixed-model ANOVA with factors PROTOCOL (Experiment 1/Experiment 2; between-subjects factor) and TIME (baseline/post-TMS; within-subjects factor). We used a significance level of $p < 0.05$. Resting state and grasping task MRI data sets were analysed in an identical way, but were not compared directly due to a categorical difference in movement artefacts (movement artefacts were larger in the grasp task than in the resting-state MRI). The severity of movement artefacts required the removal of one grasping task data set for both Experiment 1 and Experiment 2.

## Psychophysiological interaction (PPI) analysis

Psychophysiological interaction (PPI) analysis refers to the interaction between physiological activity and experimental context and thereby identifies brain areas (specifically, voxels) in which activity is more related to activity in a seed region of interest in a given experimental context. To test whether there is a change in the influence PMv (seed region) has on M1, the analysis tested for differences in the regression slope of activity in M1 on the activity in the seed region (PMv) under the experimental contexts of 'baseline' and 'post-TMS'. The change in influence of PMv on M1 can also be understood as a change in responsiveness of M1 to input from PMv. PPI analysis requires an a priori hypothesis about directionality; from physiological models it is well established that PMv provides a major input into M1 (*Dum and Strick, 2005*). Directionality of the predominant information flow from PMv to M1 was also supported by a feed-forward model validated on fMRI data acquired during performance of a grasping task (*Grol et al., 2007*) and paired-pulse TMS studies (*Davare et al., 2008*; *Buch et al., 2010*).

To test the hypothesis that repeated paired-pulse TMS stimulation of PMv and M1 altered the responsiveness of M1 to activity in PMv, we conducted a regression analysis between BOLD time series of the network nodes using Matlab R2013b (MathWorks). Individual BOLD time series for each of the two network node masks (6 mm diameter) were generated using a GLM-based design that incorporated regressors denoting potentially confounding factors such as variation in white matter (WM), grey matter (GM), and cerebrospinal fluid (CSF), and whole brain BOLD signal as implemented in FSL (fsl glm). Time series were demeaned and variance-normalised.

Analogous to statistical tests used in SBCA, we conducted paired t-tests contrasting pairwise interactions at baseline vs during post-TMS on regression coefficients (independently for Experiment 1 and Experiment 2). At a later stage, we also subjected Experiment 1 and Experiment 2 to a direct comparison by means of a mixed-model ANOVA with factors PROTOCOL (Experiment 1/Experiment 2; between-subjects factor) and TIME (baseline/post-TMS; within-subjects factor). We used a significance level of $p < 0.05$. Resting-state and grasping task MRI data sets were analysed in an identical way but were not compared directly due to a categorical difference in movement artefacts (movement artefacts were larger in the grasp task than in the resting-state MRI). The severity of movement artefacts required the removal of one grasping task data set for both the Experiment 1 and Experiment 2 condition.

## Multiple linear regression psychophysiological interaction (PPI) analysis

In analogy to a partial correlation analysis, we conducted a multiple linear regression analysis on the reaching-and-grasping network nodes (V3A, pSPL, PMd, AIP, PMv, and M1; for MNI coordinates see above in 'Regions of interest [ROI]') to understand the influence of one network node upon a specific other network node in terms of the interaction of activity in the remaining network nodes and the experimental context. Time series from the six seed masks (6 mm diameter) were generated as described above in 'Psychophysiological interaction (PPI) analysis'. To analyse the influence of a given brain area upon another, the time series of all other brain areas of interest are entered as a regressor into the multiple linear regression analysis. Statistical tests on regression coefficients were conducted as described in 'Psychophysiological interaction (PPI) analysis'.

## Dual-regression analysis (spatial regression followed by temporal regression)

To understand if co-activation patterns in large-scale networks of functional connectivity change dynamically in response to plasticity induction, we investigated networks defined by their shared spontaneous low-frequency fluctuations (<0.1 Hz). Coherence within resting-state networks (RSNs) (*Friston, 1994*) and networks during task performance (*Hampson et al., 2002*) were analysed before and after paired pulse TMS intervention using a whole-brain corrected approach. Whereas SBCA and

partial correlation analyses focused on nodes of the fronto-parietal grasping-network, this approach has the potential to identify any networks (defined as areas sharing BOLD signal temporal correlations) in which connectivity is changing as a result of the TMS intervention. This procedure was carried out completely separately for resting-state fMRI and fMRI during task performance. The approach proceeds in three stages.

To begin, concatenated multiple fMRI data sets are decomposed using ICA to identify large-scale spatial patterns of functional connectivity. We used the baseline fMRI data sets of all 23 participants who participated in this study and obtained group-averaged ICA-network maps. For seven participants who contributed to both experimental conditions, only one baseline data set was randomly selected to generate the 'group-averaged baseline' network masks; specifically, 12 'baseline' data sets were drawn from Experiment 1 and 11 'baseline' data sets were drawn from Experiment 2. By identifying ICA components based on data from both experiments, we avoided biasing our analysis as a result of any possible differences in the two groups of subjects. At the second stage, two regressions are carried out in which the ICA-derived components are regressed back against the BOLD time series from the baseline and post-plasticity induction periods in the two experiments (8 ms IPI and 500 ms IPI): firstly, to identify subject-specific temporal dynamics for each group-averaged ICA spatial component via a linear model fit (spatial regression) and, secondly, to compute subject-specific associated spatial maps, by using the generated time series as a regressor against the associated fMRI data (temporal regression). At the final stage, the resulting individual spatial component maps are collected across subjects into single 4D files (1 per original ICA network map, with the fourth dimension being subject to identification). The resulting maps—baseline and post-TMS—with one for each of the two experimental conditions, that is, Experiment 1 and Experiment 2—were then tested for voxel-wise statistical significance against the ICA maps generated from all 23 participants ('group-averaged baseline' fMRI data sets) using nonparametric permutation testing (5000 permutations) (*Nichols and Holmes, 2002*) and cluster-based thresholding and normalisation of the design matrix columns to unit standard deviation. Voxel-wise testing excluded the cerebellum. The only difference in how resting-state fMRI and grasping task fMRI were treated lay in the dimensionality estimation of ICA. The number of components was estimated automatically for both resting-state fMRI and task-positive fMRI using the Laplace approximation to the Bayesian evidence for a probabilistic principal component model (*Beckmann and Smith, 2005*), which resulted in 22 and 15 independent components for resting-state and task-positive fMRI, respectively.

## Acknowledgements

MFSR received funding from the wellcome Trust and the Medical Research Council (MRC). F-XN recieved funding from the Christopher Welch Scholarship.

## Additional information

### Funding

| Funder | Grant reference number | Author |
| --- | --- | --- |
| Wellcome Trust | | Matthew F S Rushworth |
| Medical Research Council (MRC) | | Matthew F S Rushworth |
| University Of Oxford | Christopher Welch Scholarship | Franz-Xaver Neubert |

The funders had no role in study design, data collection and interpretation, or the decision to submit the work for publication.

## Author contributions

VMJ, MFSR, Conception and design, Acquisition of data, Analysis and interpretation of data, Drafting or revising the article; F-XN, RBM, Acquisition of data, Analysis and interpretation of data, Drafting or revising the article; ERB, LV, JXO'R, Analysis and interpretation of data, Drafting or revising the article

## Ethics

Human subjects: Informed consent, including consent to publish was obtained from all subjects. The study was performed in accordance with local ethics committee approval (MKREC REF 07/Q1603/11 and Berkshire REC 11/SC/0537).

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
