## [Decision Letter]

Thank you for sending your work entitled “Causal manipulation of functional connectivity in a specific neural pathway during behaviour and at rest” for consideration at *eLife*. Your article has been favorably evaluated by Eve Marder (Senior editor), Jody Culham (Reviewing editor), and 3 external reviewers, one of whom, Alexander Sack, has agreed to share his identity.

The Reviewing editor and the reviewers discussed their comments before we reached this decision, and the Reviewing editor has assembled the following comments to help you prepare a revised submission.

All three reviewers and the Reviewing editor were very positive about the manuscript. As one reviewer stated: “This is a very clever, elegant, and informative study on how repetitive paired-pulse-induced short term synaptic efficiency changes are reflected in fMRI-based measures of functional connectivity”. Another mentioned the importance of the results for elucidating functional interactions in sensorimotor control circuits. In sum, the paper makes important methodological contributions in using fMRI to better understand and corroborate the putative effects of dual-site TMS as well as important theoretical contributions in revealing how regions within sensorimotor networks interact.

Although all involved agreed that the manuscript is well done and warrants publication in *eLife*, a number of suggestions were provided to improve the clarity of the manuscript and interpretation of the data. These points should be addressed in a revision. As per *eLife's* policy to provide clear guidelines about the changes expected in the revision, suggestions have been grouped by the main points that must be addressed in a revision and other points that should be considered. Although the points themselves are concise, where applicable, specific reviewers' comments are appended for further context.

Main points:

1) It should be made clearer (in the first paragraph of the Results section) that the paired-pulse TMS was administered at rest and the implications (vs. TMS administration during a grasping task) should be discussed.

The specific comments from Reviewer #1 provide further detail about the concerns raised:

As said by the authors, the net effect of paired-pulse TMS between PMv-M1 changes based on the task context. Using the same conditioning TMS intensity over PMv there is a net inhibition of MEPs when evoked at rest, but net facilitation when preparing a movement. The concept of 'net' effect is important because it is highly likely that both types of interactions are occurring concurrently, with only one being dominant, and visible, based on the task context. To explain these effects, it is possible that different neuronal populations are being recruited by the conditioning TMS pulse (PMv) during rest or during movement preparation because of their different excitability states in these two contexts. Therefore, assuming they have a different connectivity profile with M1, inhibition or facilitation can be seen depending on which population is recruited. This may be corroborated by the [3]. Here, different conditioning TMS (PMv) intensities were used while subjects were at rest with resulting changes in effective connectivity between PMv and M1. Again, it is likely that sub- or suprathreshold conditioning pulses recruit different neuronal populations thus leading to reveal different connectivity. To sum up, paired-TMS can show inhibition or facilitation and this may depend on which underlying neuronal population is being active.

This is where I found difficult to bring the TMS literature together with fMRI. In the present study, paired-TMS was applied always at rest, hence potentially enhancing efficacy of a single 'at rest' sub-circuit. Yet, the authors find: (23) changes in PMv-M1 fMRI connectivity during grasp preparation but not at rest; (33) changes in remote dorsolateral/dorsomedial circuits mainly at rest, but not during action, and (55) more importantly, this is even more confusing in Experiment 3. There, the authors correlate paired-TMS effect at rest or during action preparation with corresponding values of fMRI connectivity (i.e. at rest or during action preparation) but following a TMS intervention at rest, likely to potentiate synaptic efficacy of only one subtype of connections between PMv and M1.

I think it is critical the authors address these issues in the paper. In particular, how can potentiation of one sub-circuit enhance functional connectivity of a completely different one for controlling actions (see (23) above); the lack of effects in changes in the extended circuits controlling reach-to-grasp actions can be explained as only the resting-state sub-circuit has been potentiated (33); further explaining mechanisms by which one can compare effective and functional connectivity (55). This is important because the authors use one method to affect the other.

2) Further discussion should be provided regarding the use of fMRI functional connectivity to study a manipulation that causes directed effective connectivity.

Specifically, Reviewer #2 states:

The authors use paired-pulse TMS over PMv and M1 with IPI of 8ms to induce directed effective connectivity (directed influence PMv exerts over M1). This implies a temporal sequence from PMv to M1. In contrast, measures of functional connectivity during rest or task intend to capture instantaneous correlations across remote voxels. Spatial ICA is a good example of this (also used in the current paper). Isn't this than contradictory to have an increase in effective connectivity (directed influence in time from region A to B) be reflected in an increase of functional connectivity (instantaneous correlation between A and B at time point t1)? Would it not have been more straightforward to use an effective connectivity measure also in the fMRI part?

3) Relatedly, perhaps a better control condition in Experiment 2 would've been synchronous stimulation of PMv and M1 (IPI = 0)? One also wonders if the same effects would have been found if the order of stimulation had been reversed (i.e., M1 8 ms before PMv). More justification for the choice of a 500-ms IPI should be presented.

4) Because fMRI does not measure such things, the conclusions regarding “changes in synaptic efficacy” (e.g., in the first paragraph of the Discussion section) should be qualified by language that makes it clearer this is an inference.

5) As Reviewer #3 suggests (and another supports): “One of the most interesting aspects of the paper showing the competitive interaction between the dorsolateral (increased coupling) and dorsomedial (decreased coupling) circuits should be highlighted in the Abstract and Introduction sections. This finding provides new insight on how this PMv-M1 plasticity protocol induces different pathway-specific changes at a system-wide level and might further our understanding of possible interactions in the neural pathways involved in goal-directed prehension. This finding warrants further consideration in the Discussion, even if it might be hypothetical at present.”

6) The authors need to address why resting-state and grasping fMRI were conducted ∼5-15 minutes after the associative plasticity induction, given that previous PAS protocols seem to suggest maximal efficiency at about 15-30 min after the intervention and not immediately after (please see [69], Brain).

Recommended for consideration:

Some additional points were raised that may be considered at the authors' discretion:

7) Reviewer #2 suggests: “It would have been most interesting to chart the persistency of these proposed short term synaptic efficiency changes by including several post TMS fMRI session (considering they only lasted 5 minutes) and/or MEP assessments. Is there data available to still do this analysis and provide such information (e.g. in terms of duration of MEP modulation?). It would be moist revealing to show and chart that after certain duration the increase in FC decreases again to baseline.”

8) Reviewer #2 also notes: “Resting state and task fMRI data are not directly statistically compared which limits conclusions across conditions. I understand the problem of different movement artefacts. Have the authors considered normalizing the data?”

9) Reviewer #3 states: “It seems somewhat surprising that effects of PMv-M1 PAS on M1 excitability and PMv-M1 connectivity were not measured using TMS before and immediately after the associative plasticity induction during Experiment 1, rather than in a separate third experiment (especially given that this group has shown previously that this protocol lasts for at least an hour). This would provide a direct comparison between the neurophysiological index of connectivity and neuroimaging-derived index.”

10) There were mixed reviews in terms of the writing of the paper. While some thought it was well written, one reviewer thought there was room to make the manuscript clearer and less redundant. The Reviewing editor recommends the authors provide a more in-depth review of past paired associate stimulation studies and consider whether some revision would improve the flow of the manuscript.

Specifically, Reviewer #3 suggests:

A) The Introduction section would benefit from revisions to provide a more in-depth review of the background and clarify what is unique and novel in the present study. In its existing form, required background information is provided too late throughout the Results section. This makes the study motivation/rationale difficult to understand, particularly for a broader audience. Some specific topics that could better motivate and justify the current study include: a) a detailed survey of previous paired associate stimulation (PAS) studies (i.e., [69], 2002; [81]; [61]; [1]; Chao et al., 2013; [40]; etc.), specifically for a non-expert reader; b) a discussion of the differing impacts of PMv stimulation in different cognitive states, particularly for at rest versus grasp states used here; c) review of the different neural correlates of dorsolateral and dorsomedial circuits in motor control (see recent reviews on this topic by [74]; [28]); and d) a clear distinction and comparison to the group's previous work (8).

B) The Results could be reduced greatly (by ∼25%), and substantial re-writing of the text is necessary not only to improve the narrative, but also to remove redundancy (i.e., the first eight paragraphs of the Results section could easily be combined with text from Introduction section and Results section for Experiment 1). In addition, findings from the control experiment (Experiment 2; in the subsection “Experiment 2: paired stimulation of PMv and M1 at 500ms IPI” of the Results) could easily be combined with the text in the Results section of Experiment 1 to provide a direct contrast between the main experiment and control. This would further bolster the narrative.

11) The Reviewing editor wondered why functional connectivity between the dorsomedial and dorsolateral subnetworks (esp. PMv-PMd) was not investigated, especially considering that PMd may play a more important role in grasp programming than the standard two-substreams connections suggest (e.g., Raos, 2004, J Neurophysiol).

---

## [Author Response]

*1) It should be made clearer (in the first paragraph of the Results section) that the paired-pulse TMS was administered at rest and the implications (vs. TMS administration during a grasping task) should be discussed*.

*The specific comments from Reviewer #1 provide further detail about the concerns raised*:

*As said by the authors, the net effect of paired-pulse TMS between PMv-M1 changes based on the task context. Using the same conditioning TMS intensity over PMv there is a net inhibition of MEPs when evoked at rest, but net facilitation when preparing a movement. The concept of 'net' effect is important because it is highly likely that both types of interactions are occurring concurrently, with only one being dominant, and visible, based on the task context. To explain these effects, it is possible that different neuronal populations are being recruited by the conditioning TMS pulse (PMv) during rest or during movement preparation because of their different excitability states in these two contexts. Therefore, assuming they have a different connectivity profile with M1, inhibition or facilitation can be seen depending on which population is recruited. This may be corroborated by the*
[3]*. Here, different conditioning TMS (PMv) intensities were used while subjects were at rest with resulting changes in effective connectivity between PMv and M1. Again, it is likely that sub- or suprathreshold conditioning pulses recruit different neuronal populations thus leading to reveal different connectivity. To sum up, paired-TMS can show inhibition or facilitation and this may depend on which underlying neuronal population is being active*.

We agree with the reviewer that different neuronal subpopulations will drive excitatory or inhibitory patterns of PMv-M1 interactions. It is important to remember, however, that the projections from premotor cortex to M1, like many inter-regional connections, are mainly of a glutamatergic (excitatory) nature but within M1 they can either synapse directly onto pyramidal cells (facilitatory influence; less common) or onto inhibitory interneurons (majority) ([72], Cereb Cortex). The cognitive state may determine the weight of either connection or else it may determine activity levels in the inhibitory neurons via connections from areas other than premotor cortex. ([3], Clin Neurophysiol) did indeed show different patterns of connectivity between PMv and M1 in response to varying conditioning stimulus intensities and an inhibitory effect of PMv on M1 was demonstrated for a PMv conditioning pulse with an intensity of 110% of resting motor threshold at rest (see also: [8], J Neurosci). Importantly, we also chose a PMv conditioning pulse of 110% RMT and we concur with Baumer and colleagues in finding that it predominantly causes inhibition of M1-induced MEPs at rest. However, just like the reviewer, we agree that influences other than inhibitory ones are likely to be at play. In line with the reviewer’s suspicion we have reported that facilitation of MEPs during the grasping task is also seen ([8], J Neurosci). In line with the editorial guidance we received we have noted this fact at the beginning of the Results section where we have cited the work of Baumer and colleagues. We have written the following:

“In Experiment 1 each participant (N=15) underwent two sets of two 5-minute fMRI scans for the purpose of assessing functional connectivity in both a baseline state […] We counterbalanced the order of baseline and post-TMS scans across subjects (half of the subjects had the post-TMS scan before the baseline scan on two different days; Figure 1).”

*This is where I found difficult to bring the TMS literature together with fMRI. In the present study, paired-TMS was applied always at rest, hence potentially enhancing efficacy of a single 'at rest' sub-circuit. Yet, the authors find: (*[23]*) changes in PMv-M1 fMRI connectivity during grasp preparation but not at rest; (*[33]*) changes in remote dorsolateral/dorsomedial circuits mainly at rest, but not during action, and (*[55]*) more importantly, this is even more confusing in Experiment 3. There, the authors correlate paired-TMS effect at rest or during action preparation with corresponding values of fMRI connectivity (i.e. at rest or during action preparation) but following a TMS intervention at rest, likely to potentiate synaptic efficacy of only one subtype of connections between PMv and M1*.

We would like to emphasise that we are very grateful to the reviewer for the time that he/she has spent in reviewing our manuscript and for the very wise comments they have made. However we think that the reviewer’s comments in this paragraph are not quite correct but instead the comments in the reviewer’s previous paragraph were more important. In the present paragraph the reviewer almost appears to argue that we should expect that different connections are stimulated when subjects are at rest or in a task state, but as we have already explained in the previous paragraph we know that there is only one main type of connection between premotor cortex and M1, a glutamatergic one that synapses on both pyramidal neurons and inhibitory interneurons. This means that there is no “'at rest' sub-circuit” within PMv and M1 or no “subtypes of connection” from PMv to M1; there rather are excitatory connections between PMv and M1 that have a variety of targets within M1. Strengthening this single type of connection while subjects are at rest is already known to lead to subsequent increments in facilitatory influences of PMv over M1 when subjects are subsequently engaged in a task and to subsequent increments in inhibitory influences of PMv over M1 when subjects are subsequently at rest ([8], J Neurosci). Again, the reviewer was surely correct in their previous paragraph when they suggested that it is just that different effects of stimulation are more visible or detectable, using the very indirect indices of neural activity furnished by fMRI and TMS, depending on subjects’ cognitive state. Our present fMRI results can be interpreted from the same perspective; some changes in connectivity are simply more visible when subjects are at rest.

It addition, it is important to remember that the induction of increased excitatory connection strength between some parts of a circuit is expected to be associated with decreased connection strength in other parts of the circuits. This can be seen, for example, in the work of Vogels and colleagues on inhibitory anti-memories ([77], Science), but again the sensitivity that fMRI will have for detecting such changes may depend on the task context. While the occurrence of reduction in connectivity in other parts of the motor circuit is a novel result (that is why we are submitting the results to *eLife*), it is a result that has been predicted on theoretical grounds.

*I think it is critical the authors address these issues in the paper. In particular, how can potentiation of one sub-circuit enhance functional connectivity of a completely different one for controlling actions (see (*[23]*) above); the lack of effects in changes in the extended circuits controlling reach-to-grasp actions can be explained as only the resting-state sub-circuit has been potentiated (*[33]*); further explaining mechanisms by which one can compare effective and functional connectivity (*[55]*). This is important because the authors use one method to affect the other*.

It is unlikely that the PMv-M1 neural circuits which are recruited at rest in contrast to those recruited during grasping are completely independent. In fact, as explained in the previous paragraph, the evidence to date suggests precisely the opposite to be the case. Strengthening the single type of excitatory connection that exists between PMv and M1 (which synapses on both pyramidal neurons and inhibitory interneurons) while subjects are at rest is already known to lead to subsequent increments in facilitatory influences of PMv over M1 when subjects are subsequently engaged in a task and to subsequent increments in inhibitory influences of PMv over M1 when subjects are subsequently at rest ([8], J Neurosci). Moreover, excitation of a smaller volume of neural tissue by means of paired microwire stimulation (PMv-M1) in macaques has also been shown to exert both facilitatory and inhibitory effects on corticospinal outputs that change in weight with different cognitive states, anaesthetised and behaving ([60], J Physiol). Our previous studies and other studies investigating PMv-M1 connectivity have shown that a change in behavioural context is sufficient to cause a complete reversal of the influence exerted by PMv over M1 ([7], J Neurosci; [8], J Neurosci; [13], J Physiol), even in the absence of a change in stimulation parameters. The change from facilitation to inhibition is driven by a shift in subpopulation weights and unlikely to be driven by independent PMv-M1 circuits. In order to pre-empt such misunderstanding we have, in the revised manuscript, explained this point in greater detail in the Introduction of our manuscript.

“As is the case for other inter-regional connections, the connections between premotor cortex and M1 are glutamatergic, excitatory ones, but there are synapses on both pyramidal neurons and inhibitory interneurons within M1. […] PMv microwire stimulation in macaques has also been shown to exert both facilitatory and inhibitory effects on corticospinal outputs as a function of the animal’s state.”

*2) Further discussion should be provided regarding the use of fMRI functional connectivity to study a manipulation that causes directed effective connectivity*.

*Specifically, Reviewer #2 states*:

The authors use paired-pulse TMS over PMv and M1 with IPI of 8ms to induce directed effective connectivity (directed influence PMv exerts over M1). This implies a temporal sequence from PMv to M1. In contrast, measures of functional connectivity during rest or task intend to capture instantaneous correlations across remote voxels. Spatial ICA is a good example of this (also used in the current paper). Isn't this than contradictory to have an increase in effective connectivity (directed influence in time from region A to B) be reflected in an increase of functional connectivity (instantaneous correlation between A and B at time point t1)? Would it not have been more straightforward to use an effective connectivity measure also in the fMRI part?

First, we think it is important to study functional connectivity with functional magnetic resonance imaging (fMRI) because this is the measure most neuroscientists including clinicians use to study the interactions between brain regions. Moreover, we note that functional connectivity is the more general process, a change in effective connectivity will always be reflected in a change in functional connectivity, but the opposite is not always the case. Second, we note that in the present investigation we can only analyze block-wise changes in connectivity; we used a block-based design in our experiment (before versus after plasticity induction) because it was the first time anybody had ever conducted an experiment of this sort (that is why we have sent the manuscript to *eLife*) and we wanted to have the high power that a block-based design confers. Unfortunately, this means we have to disregard temporal information as a factor of interest in the fMRI analysis. It is unlikely that the temporal resolution of fMRI, particularly in the block-based design we used, would enable determination of whether or not coupling between brain areas is occurring exactly instantaneously or is occurring with an 8 ms lag. Instead such questions of directional influences of one brain area over another are perhaps best determined by probing connectivity with a very high temporal resolution technique such as further pulses of TMS ([8], J Neurosci). Such an approach, however, while providing precise temporal information, would not provide the more detailed spatial and anatomical information about coupling changes we were able to obtain in the present investigation.

*3) Relatedly, perhaps a better control condition in Experiment 2 would've been synchronous stimulation of PMv and M1 (IPI = 0)? One also wonders if the same effects would have been found if the order of stimulation had been reversed (i.e., M1 8 ms before PMv). More justification for the choice of a 500-ms IPI should be presented*.

For the control condition we chose an interpulse interval (IPI) which is unlikely to evoke changes in pathway connectivity but which still meant stimulation involved the same number of pulses at the exact same frequency and in the same order. We decided: 1) against reversing the order of conditioning and test stimulus as we have already demonstrated that this stimulation order leads to LTD (as assessed—albeit not with fMRI but—by examining the impact of further PMv TMS pulses on M1 ([8], J Neurosci); 2) against stimulating both areas at the same time because I-wave interactions are likely to occur at such an IPI ([60], J Physiol); 3) against any time interval below 50 ms as there is evidence of plasticity induction at such intervals within the motor system of freely behaving monkeys ([35], Nature). Moreover we noted that long-interval intracortical inhibition (LICI) within M1 has been demonstrated with TMS using IPIs of up to 200 ms ([75], Electroencephalogr Clin Neurophysiol). Therefore, we chose an IPI which is unlikely to induce spike timing dependent plasticity; other intervals in the hundred milliseconds range might equally have been chosen.

We added the following to the Introduction section:

“We decided on a 500 ms IPI for the control condition following the exclusion of several other alternative IPIs; […] Admittedly, other intervals in the hundred milliseconds range might equally have been chosen.”

*4) Because fMRI does not measure such things, the conclusions regarding “changes in synaptic efficacy” (e.g., in the first paragraph of the Discussion section) should be qualified by language that makes it clearer this is an inference*.

We agree with the reviewer that we do not measure synaptic efficacy directly, but we have shown that using a protocol known to change synaptic efficacy does result in the predicted changes ([8], J Neurosci). Therefore, we will clarify our statement in the Discussion that makes this train of inference explicit:

“In this study, we describe the functional connectivity signature in fMRI data of short-term synaptic potentiation within a specific anatomical pathway. […] From our experiments we infer that functional connectivity is not only shaped by structural connections but also by short-term plastic changes in synaptic efficacy.”

*5) As Reviewer #3 suggests (and another supports): “One of the most interesting aspects of the paper showing the competitive interaction between the dorsolateral (increased coupling) and dorsomedial (decreased coupling) circuits should be highlighted in the Abstract and Introduction sections. This finding provides new insight on how this PMv-M1 plasticity protocol induces different pathway-specific changes at a system-wide level and might further our understanding of possible interactions in the neural pathways involved in goal-directed prehension. This finding warrants further consideration in the Discussion, even if it might be hypothetical at present*.*”*

We thank the reviewer for highlighting this result. We agree that this is one of the most exciting results. It demonstrates that strengthening interregional connectivity between two areas in a pattern consistent with Hebbian plasticity does not only lead to increased functional connectivity between these two regions; most notably it also affects coupling between areas that have not been targeted by stimulation.

Several recent studies have shown that lesions or temporary disruption of brain areas as well as lesions to connections between brain areas can affect distant areas and connections (O’Shea et al, Neuron, 2007; Hartwigsen et al, J Neurosci, 2012; O’Reilly et al, PNAS, 2013). These changes are thought to be partly compensatory. For example, in the study by O’Shea and colleagues (2007) it is suggested that “activity” in contralateral “non-dominant” PMd is increased after disruption of ipsilateral PMd. This enhancement of contralateral PMd is accompanied by preserved performance in a conditional action selection task. Similarly, [34] show that action reprogramming can be preserved after PMd interference if the supramarginal gyrus is uncompromised. This study suggests a rapid redistribution of functional weights in order to compensate for interference. Moreover it has been shown that the interruption of specific pathways has effects far beyond the regions that are directly connected by the pathway (O’Reilly et al., PNAS, 2013).

Here by contrast, we study functional enhancement of a pathway rather than interruption of a region or pathway. We show that plasticity-induction in a given connection also affects distant pathways. The increase in functional coupling between PMv and AIP following the TMS-protocol is accompanied by a decrease in functional connectivity between PMd and M1, as well as between PMd and V6A. It is possible that these accessory decreases in distant connections fulfil similar “compensatory” roles. Inhibitory plasticity might accompany excitatory plasticity in order to stabilise neural networks involved in learning (Vogels et al., Science, 2011). Vogels et al offer a solution to the question of how the careful balance of excitatory and inhibitory inputs into a cortical neuron could self-organize within the activity dynamics of neuronal networks. They suggest that inhibitory spike timing-dependent plasticity could balance excitatory inputs. Learning or the formation of associative excitatory-excitatory connections in such networks would require the co-adaptation of excitatory and inhibitory synapses. Although Vogels’ and colleagues’ ideas largely make predictions about structural and functional properties of local neural circuits, the results of this experiment could be taken to suggest that similar principles apply to the network and systems level.

Additionally enhancement of one pathway might be accompanied by diminution of a parallel pathway if both of them compete for their influence on a particular target structure such as M1. It has been argued that two pathways for movement preparation—the dorsomedial visuomotor stream (V6A – PMd) and the dorsolateral visuomotor stream (AIP – PMv)—complement each other by driving movement selection proportional to the amount of information available in each stream (Verhagen et al., J Neurosci, 2008). It remains to be determined how exactly movement selection is biased towards dorsolateral or dorsomedial streams and whether this is a categorical or a gradient process. The study of multi-sensory integration has generated some theoretical constructs for how integration of information from two different streams might be achieved (Ernst and Banks, Nature, 2002). In this framework different inputs are combined according to maximum likelihood estimation. Hence a channel providing more reliable information for example because of higher signal-to-noise ratio might then be guiding movement selection more strongly with the weights being inversely proportional to the variance of the signal from the respective channels. If the dorsomedial and dorsolateral visuomotor pathways compete for guiding motor control then enhancement of information processing in one of the pathways might suppress the other stream.

Future research needs to understand the relation of these different pathways and how they interact and potentially compete for guiding movement selection. With more detailed knowledge about the structural skeleton and the functional relationship of these streams we might be able to predict the complex effects of learning and plasticity not only on the particular network primarily involved in learning and plasticity but also on other parallel streams and networks. More generally this might eventually contribute to a better understanding of network effects relating to learning, development and degeneration (Dayan and Cohen, Neuron, 2011; Fair et al, PNAS, 2007; Seeley et al, Neuron., 2009). For this line of research whole-brain approaches such as fMRI or MEG might prove advantageous over only looking at local effects of plasticity, such as changes in the MEP (Buch, Johnen et al., 2011).

As suggested by the reviewer we have made the following changes:

In the Abstract:

“Moreover, we show that strengthening connectivity between these nodes has effects on a wider network of areas, such as decreasing coupling in a parallel motor programming stream.”

In the Introduction:

“Recent studies have shown that lesions and disruption of brain areas as well as lesions to connections between brain areas can affect distant areas and connections. […] Here by contrast, we study the functional enhancement of a pathway rather than the disruption of a region or pathway and its effect on coupling within and outside the targeted network.”

In the Discussion:

“The exact functional role of these accessory decreases in functional coupling in distant connections requires further investigation. […] For this line of research whole-brain approaches such as fMRI or MEG might have some advantages in some contexts in comparison to examining more local effects of plasticity, such as changes in MEPs.”

*6) The authors need to address why resting-state and grasping fMRI were conducted ∼5-15 minutes after the associative plasticity induction, given that previous PAS protocols seem to suggest maximal efficiency at about 15-30 min after the intervention and not immediately after (please see*
[69]*, Brain)*.

Relating to the study of [69] it is true that visual inspection of PAS induced changes in corticospinal output might suggest that the maximal efficacy was not achieved before 15min after plasticity induction; however, the effects were not statistically different between early and late points following intervention.

As for our study, the results from our previous study (2011) did not show any differences between efficacy immediately after intervention in comparison to +30 min or +60 min post-intervention. We therefore have no evidence for our ppTMS protocol that the effects varied in efficiency within the first 60 min following interventions.

We now refer to this issue in the Methods section:

“TMS was applied outside the MRI scanner room. Participants walked to the MRI scanner and scanning commenced within three to four minutes. Note, previous neurophysiological experiments suggest plasticity induction should last at least one hour with this protocol and that there were no differences in efficacy immediately after intervention in comparison to +30 min or +60 min post-intervention.”

*Recommended for consideration*:

*Some additional points were raised that may be considered at the authors' discretion*:

*7) Reviewer #2 suggests: “It would have been most interesting to chart the persistency of these proposed short term synaptic efficiency changes by including several post TMS fMRI session (considering they only lasted 5 minutes) and/or MEP assessments. Is there data available to still do this analysis and provide such information (e.g. in terms of duration of MEP modulation?). It would be moist revealing to show and chart that after certain duration the increase in FC decreases again to baseline*.*”*

It is true that the fMRI sessions were relatively short (although not quite as short as the reviewer envisages because data from both task and resting state conditions were collected). Nevertheless collecting the data required the rapidly successive use of several pieces of equipment and procedures including electromyography, neuronavigation, transcranial magnetic stimulation, use of bespoke grasping apparatus in the magnetic resonance imaging environment, and functional magnetic resonance imaging. All of this was extremely challenging in a busy research hospital environment. We would, however, like to draw attention to the results published as Buch, Johnen et al. (2011) where we demonstrated prolonged changes in synaptic efficacy for more than one hour. The measurements of synaptic efficacy in that study were, however, obtained in a very different way. Instead of measuring whole brain activity coupling with fMRI we recorded the impact of the plasticity induction protocol on subsequent PMv TMS pulses on M1. We showed that their effect was enhanced for more than one hour after plasticity induction. We have noted this in the revised manuscript (see below). For the rest, we agree that it would have been very interesting to track the effects over multiple post-fMRI sessions. For the current study, we focussed our design and analysis on the first time points following intervention to maximise power for the pre-post analysis. We are planning to address the point of longevity of changes in functional connectivity in future studies.

The following changes were made to the Discussion and Methods sections:

“In the current study we did not track the duration of these changes in functional coupling after the intervention. However we note that in a previous study changes in effective connectivity were shown to last more than one hour.”

“Note, previous neurophysiological experiments suggest plasticity induction should last at least one hour with this protocol and that there were no differences in efficacy immediately after intervention in comparison to +30 min or +60 min post-intervention.”

*8) Reviewer #2 also notes: “Resting state and task fMRI data are not directly statistically compared which limits conclusions across conditions*. *I understand the problem of different movement artefacts. Have the authors considered normalizing the data?”*

We have tried different kinds normalization of the resting state data, but felt this comparison was not appropriate. Also, we note that the experiment was explicitly designed with the goal of separately analysing the two task periods and maximizing power for the pre-post analyses, hence the block-wise manipulation.

*9) Reviewer #3 states: “It seems somewhat surprising that effects of PMv-M1 PAS on M1 excitability and PMv-M1 connectivity were not measured using TMS before and immediately after the associative plasticity induction during Experiment 1, rather than in a separate third experiment (especially given that this group has shown previously that this protocol lasts for at least an hour). This would provide a direct comparison between the neurophysiological index of connectivity and neuroimaging-derived index*.*”*

We appreciate the reviewers’ exciting suggestions for follow-ups and ways to further improve our design. It is true that our analysis is based on a between-session analysis, although we have demonstrated extremely high test-retest reliability in paired-pulse paradigms, which makes us confident our results are reliable. Given the logistical difficulties of this type of study (electromyography, neuronavigation, transcranial magnetic stimulation, use of bespoke grasping apparatus in the magnetic resonance imaging environment, and functional magnetic resonance imaging) we opted for this setup to maximize the chance of getting good and reliable data.

*10) There were mixed reviews in terms of the writing of the paper. While some thought it was well written, one reviewer thought there was room to make the manuscript clearer and less redundant. The Reviewing editor recommends the authors provide a more in-depth review of past paired associate stimulation studies and consider whether some revision would improve the flow of the manuscript*.

*Specifically, Reviewer #3 suggests*:

*A) The Introduction section would benefit from revisions to provide a more in-depth review of the background and clarify what is unique and novel in the present study. In its existing form, required background information is provided too late throughout the Results section. This makes the study motivation/rationale difficult to understand, particularly for a broader audience. Some specific topics that could better motivate and justify the current study include: a) a detailed survey of previous paired associate stimulation (PAS) studies (i.e.,*
[69]*, 2002;*
[81]*;*
[61]*;*
[1]*; Chao et al., 2013;*
[40]*; etc.), specifically for a non-expert reader; b) a discussion of the differing impacts of PMv stimulation in different cognitive states, particularly for at rest versus grasp states used here; c) review of the different neural correlates of dorsolateral and dorsomedial circuits in motor control (see recent reviews on this topic by*
[74]*;*
[28]*); and d) a clear distinction and comparison to the group's previous work (*[8]*)*.

We thank the reviewer for this comment and have improved the introduction to make is easier to understand for a broader audience and also emphasised our motivation for the current study. We added a more detailed survey of PAS studies in the Introduction section:

“Several TMS protocols have been shown to induce changes in excitability in primary motor cortex (M1) using repetitive stimulation of M1 itself or stimulation of premotor regions projecting to M1. […] Investigations that applied paired-pulse TMS over interconnected sites—for example, homotopical M1 sites, M1 and the supplementary motor area (SMA), and M1 and posterior parietal cortex—demonstrated altered motor cortical excitability.”

Furthermore, we explained the different impacts of PMv stimulation in relation to different cognitive states, in the Introduction section:

“As is the case for other inter-regional connections, the connections between premotor cortex and M1 are glutamatergic, excitatory ones, but there are synapses on both pyramidal neurons and inhibitory interneurons within M1. […] PMv microwire stimulation in macaques has also been shown to exert both facilitatory and inhibitory effects on corticospinal outputs as a function of the animal’s state.”

We also shortly reviewed the different neural correlates of the dorsolateral and dorsomedial circuits in motor control:

“PMv and M1 are a part of the so-called “dorsolateral circuit“ of areas composed of the anterior intraparietal (AIP) area, areas PF and PFG in the inferior parietal lobule, and PMv and M1 in the frontal lobes. During complex motor behaviour such as reaching and grasping this dorsolateral sensorimotor stream is complemented by a “dorsomedial circuit“ composed of dorsal premotor (PMd), medial intraparietal area (MIP) and posterior superior parietal cortex (pSPL).”

In line with the reviewer’s suggestion, we emphasised the novel aspect of our work, with a particular focus on how our study goes beyond previous TMS manipulations inducing compensatory plasticity, also in the Introduction:

“Recent studies have shown that lesions and disruption of brain areas as well as lesions to connections between brain areas can affect distant areas and connections. […] Here by contrast, we study the functional enhancement of a pathway rather than the disruption of a region or pathway and its effect on coupling within and outside the targeted network.”

*B) The Results could be reduced greatly (by ∼25%), and substantial re-writing of the text is necessary not only to improve the narrative, but also to remove redundancy (i.e., the first eight paragraphs of the Results section could easily be combined with text from Introduction section and Results section for Experiment 1). In addition, findings from the control experiment (Experiment 2; in the subsection “Experiment 2: paired stimulation of PMv and M1 at 500ms IPI” of the Results) could easily be combined with the text in the Results section of Experiment 1 to provide a direct contrast between the main experiment and control. This would further bolster the narrative*.

We apologize for the redundancy. We noticed that the reviewer is correct that some results regarding the comparison of experiments 1 and 2 were indeed present twice. We have removed repetitions and we have retained comparison of the two experiments in the section of the manuscript that details the results of experiment 2. We were able to identify the relevant sections. We have followed the suggestions and substantially re-written the results section mentioned by the reviewer; for a better understanding, we also added a table to the manuscript which presents the statistical results of the different analyses (Table 2). If the addition of the table is not in agreement with the reviewers, we are happy to remove the table.

The following is an excerpt of the changes made to different parts of the manuscript:

“Using a higher-level analysis (mixed-model ANOVA) with between-subjects factor “PROTOCOL” we directly contrasted the effects from Experiment 1 and Experiment 2 for each of the analyses conducted. […] A partial correlation analysis contrasting Experiment 1 with Experiment 2 confirmed that during task, PMv-M1 coupling was only changed in the grasping condition following plasticity induction with an 8ms IPI (mixed-model ANOVA: TIME by PROTOCOL interaction: F(1,26)=7.47, P=0.011; Experiment 2 during task: paired t-test: t(13)=1.18, P=0.26).”

“We namely found no changes in pairwise coupling in Experiment 2 (Experiment 2 at rest: paired t-tests: AIP-PMv: t(14)=0.41, P=0.96; pSPL-PMd: t(14)=-1.18, P=0.26; PMd-M1: t(14)=0.01, P=0.99). When contrasting pairwise interactions from both experiments, we confirmed that increases in AIP-PMv connectivity only occurred following STDP (mixed-model ANOVA: TIME by PROTOCOL interaction: AIP-PMv: F(1,28)=5.74, P=0.024) with a further decrease in PMd-M1 connectivity (mixed-model ANOVA: TIME by PROTOCOL interaction: PMd-M1: F(1,28)=4.44, P=0.044) and a tendency for the decrease in pSPL-PMd connectivity (mixed-model ANOVA: TIME by PROTOCOL interaction: pSPL-PMd: F(1,28)=3.66, P=0.066).”

*11) The Reviewing editor wondered why functional connectivity between the dorsomedial and dorsolateral subnetworks (esp. PMv-PMd) was not investigated, especially considering that PMd may play a more important role in grasp programming than the standard two-substreams connections suggest (e.g., Raos, 2004, J Neurophysiol)*.

In retrospect, the suggestion of examining PMv-PMd coupling seems like a sensible one. However, if there were some sort of effect it ought to have been apparent in the ICA. Even when we relax the statistical criterion we cannot see evidence of a PMd-PMv coupling change. We are not sure of the reasons and are cautious about over-emphasizing a negative result. We have, therefore, left this point undiscussed in the revised manuscript but we are happy to make further changes upon request.